# Effectiveness of participatory community solutions strategy on improving household and provider health care behaviors and practices: A mixed-method evaluation

**Gizachew Tadele Tiruneh**[1]*, **Nebreed Fesseha Zemichael**[1], **Wuleta Aklilu Betemariam**[1], **Ali Mehryar Karim**[2]*

1 The Last Ten Kilometers (L10K) 2020 Project, JSI Research & Training Institute, Inc., Addis Ababa, Ethiopia, 2 Bill & Melinda Gates Foundation, Addis Ababa, Ethiopia

* gizachew_tadele@et.jsi.com (GTT); ali_karim@jsi.com (AMK)

## Abstract

### Introduction

We implemented a participatory quality improvement strategy in eight primary health care units of Ethiopia to improve use and quality of maternal and newborn health services.

### Methods

We evaluated the effects of this strategy using mixed-methods research. We used before-and-after (March 2016 and November 2017) cross-sectional surveys of women who had children 0–11 months to compare changes in maternal and newborn health care indicators in the 39 communities that received the intervention and the 148 communities that did not. We used propensity scores to match the intervention with the comparison communities at baseline and difference-in-difference analyses to estimate intervention effects. The qualitative method included 51 in-depth interviews of community volunteers, health extension workers, health center directors and staff, and project specialists.

### Results

The difference-in-difference analyses indicated that 7.9 percentage points (95% confidence interval [CI]: 1.8–13.9%) increase in receiving skilled delivery care between baseline and follow-up surveys in the intervention area that is attributable to the strategy. The intervention effect on postnatal care in 48 hours of the mother was 15.3% (95% CI: 7.4–23.2). However, there was no evidence that the strategy affected the seven other maternal and newborn health care indicators considered. Interview participants said that the participatory design and implementation strategy helped them to realize gaps, identify real problems, and design appropriate solutions, and created a sense of ownership and shared responsibility for implementing interventions.

**Data Availability Statement:** All relevant data are within the manuscript and its Supporting Information files.

**Funding:** The article write-up and publication fee was supported by the Bill & Melinda Gates Foundation, Grant Number OPP1131042. JSI Research & Training Institute, Inc. has provided us support in the form of salaries for authors [GT, AK, NZ, WB]. However, any of the funders did not have role in study design, data collection and analysis, decision to publish, or preparation of the manuscript."

**Competing interests:** The authors declare that they have no competing interests. The authors have been working for JSI Research & Training Institute, Inc., a commercial company. We declared that this commercial affiliation does not alter our adherence to PLOS ONE policies on sharing data and materials. The senior author of this manuscript has been working for JSI Research and Training Institute Inc. during the design, field work, and report-up of this manuscript. However, he recently left JSI and joined Gates Foundation. We would like to declare that we do not have any conflict of interest with the Foundation-paid staff in preparing this manuscript.

**Abbreviations:** ANC, antenatal care; BEmONC, basic emergency obstetric and newborn care; CBDDM, community-based data for decision making; EmONC, emergency obstetric and newborn care; FMOH, Federal Ministry of Health; HEP, Health Extension Program; HEW, health extension worker; HSTP, Health Sector Transformation Plan; IDI, in-depth interview; KMC, kangaroo mother care; L10K, Last Ten Kilometers 2020 project; MgSO4, magnesium sulfate; MNH, maternal and newborn health; PHCU, primary health care unit; PNC, postnatal care; PROM, prolonged premature rupture of membrane; PSBI, possible serious bacterial infection; QI, quality improvement; RMNCH, reproductive, maternal, newborn, and child health; SDG, Sustainable Development Goal; SNNP, Southern Nations, Nationalities and Peoples; WDA, women's development army.

## Conclusions

Community participation in planning and monitoring maternal and newborn health service delivery improves use of some high-impact maternal and newborn health services. The study supports the notion that participatory community strategies should be considered to foster community-responsive health systems.

## Introduction

There is a wealth of evidence on a range of essential interventions to prevent maternal and newborn deaths [1–3]. More than half of all newborn deaths could be averted by providing care during the postpartum period: 30% can be averted with care of small and ill newborns; 12% with care of healthy newborns; and 10% with immediate newborn care [2]. Evidence also suggests that high coverage and quality of essential packages for maternal and newborn health (MNH) services, basic and emergency obstetric care, and postnatal care across the continuum of care could avert about two-thirds of newborn and child deaths [4]. A recent systematic review also shows the continuity of care from antepartum to postpartum periods may reduce the risk of combined newborn, perinatal, and maternal mortality by 15% [5] and reduce neo-natal and perinatal mortality risk by 21% and 16%, respectively [6].

However, such life-saving care has not been implemented adequately in low-and-middle-income countries [6–9]. The Ethiopian situation is not different. Despite a three-fold increase in institutional delivery in Ethiopia, from 10% in 2011 to 28% in 2016, there are high dropout rates across the continuum of MNH care [10, 11]. The Ethiopian Demographic and Health Survey 2016 reports that while 62% of pregnant women received skilled antenatal care (ANC) and 28% delivered their babies under the guidance of a skilled person, only 17% received post-natal care (PNC) within 48 hours [10, 11]. Moreover, studies report that some key components are not provided at ANC, delivery, and PNC stages [12–14], even if women visit health facilities. The proportion of women with obstetric complications treated in health facilities (i.e., who met need for emergency obstetric and newborn care [EmONC]) remains very low, though it did increase from 6% in 2008 to 18% in 2016 [15].

Evidence shows that use of MNH services is influenced by myriad socio-demographic, health service-related, and cultural factors [16–26]. Multiple studies find maternal age, parity, lack of time, education, marital status, women's economic status, residence, and distance to health facility are significant predictors of use of maternity care services [16, 18, 27, 28]. Accordingly, intervention strategies targeting improved use of MNH services should address both supply- and demand-side factors.

Community participation has been promoted as a critical component of a human rights-based approach to primary health care to build a resilient health system [29, 30]. It has been presumed to improve health outcomes, access, equity, acceptability, service quality, and responsiveness [31]. Community participation in MNH program planning and implementation as well as in quality improvement (QI) processes for MNH services is a compelling strategy for improving service quality [30, 32]. However, its impact on QI depends on its proper design of strategy and implementation.

Evidence from developing countries shows that community participation is effective in decreasing newborn mortality [33–35]. Studies also indicate that community participation in health care QI strategies lowers newborn mortality rates. In Malawi, a randomized controlled trial of combined participatory women's groups and QI at health centers demonstrates a

reduction in newborn mortality [36]. Another observational study in Ethiopia indicated that QI approaches increase use of maternal care services [37]. There is, however, no sufficient evidence that community participation is associated with better maternal health outcomes, such as improving service access, use, quality, or responsiveness [33, 34]. Particularly, the effect of community-participation on the uptake of skilled care and immediate postnatal care is not well documented [33, 37–39].

We implemented a participatory community and facility QI intervention, the Participatory Community Solutions (PC-Solutions) strategy to remove supply- and demand-side barriers to desirable MNH care behaviors and practices. This study used mixed-methods research to evaluate the effect of the PC-Solutions strategy on improving MNH care behaviors and practices in selected rural areas of Ethiopia.

In the first phase, the quantitative surveys were conducted to evaluate the effectiveness of the PC-Solution strategy on the use of MNH behaviors and practices. In the second phase, qualitative research was conducted to explain the findings of the quantitative study and facility surveys; document what happened during the PC-Solutions implementation process; understand the complex participatory community QI process; and assess the scalability of the intervention. The qualitative component was to illuminate how the intervention affected outcomes.

## Methods

### Study settings

**Ethiopian health system.** Ethiopia has a three-tiered health system designed to deliver services. The first level is primary health care, which serves an administrative district (*woreda*) with an average population of about 100K. The second level is a general hospital that serves the catchment population of approximately 10 woredas (about 1 million people), and the third level is a specialized hospital that serves the catchment population of five general hospitals (about 5 million people). In rural woredas, primary care comprises a primary hospital and a primary health care unit (PHCU) of one health center and five satellite health posts for every 25K people in the woreda [40].

Within the PHCU is Ethiopia's flagship Health Extension Program (HEP), which establishes one health post and deploys two health extension workers (HEWs) in each *kebele* (a community of about 5,000 people) in the country to provide basic promotive, preventive, and curative health services [41]. To extend the reach of the HEP and mobilize the community and households, the Federal Ministry of Health (FMOH) established a network of women's development army (WDA) members. Each WDA member is assigned to five households and encourages families to adopt and practice healthy behaviors [42].

The FMOH is committed to achieving the health-related Sustainable Development Goals (SDGs). The targets for improving reproductive, maternal, newborn, and child health (RMNCH) outcomes of its Health Sector Transformation Plan (HSTP) are aligned with the SDGs of reducing maternal and newborn mortality rates [43, 44]. To achieve these objectives, the HSTP aims to strengthen health systems to provide universal access to high-quality promotive, preventive, curative, and rehabilitative services. One of the key aspects of ensuring universal access to health services is engaging communities in the governance and accountability of their health system by monitoring program performance and ensuring service quality [45].

**Project description.** JSI Research & Training Institute, Inc. (JSI), with funding from the Bill & Melinda Gates Foundation, implements the Last Ten Kilometers (L10K) 2020 project strategies to involve communities in improving high-impact RMNCH care behavior and practices in 115 woredas in four of the most populous regions of the country (Amhara, Oromia, Southern Nations, Nationalities and Peoples [SNNP], and Tigray), covering about 19 million

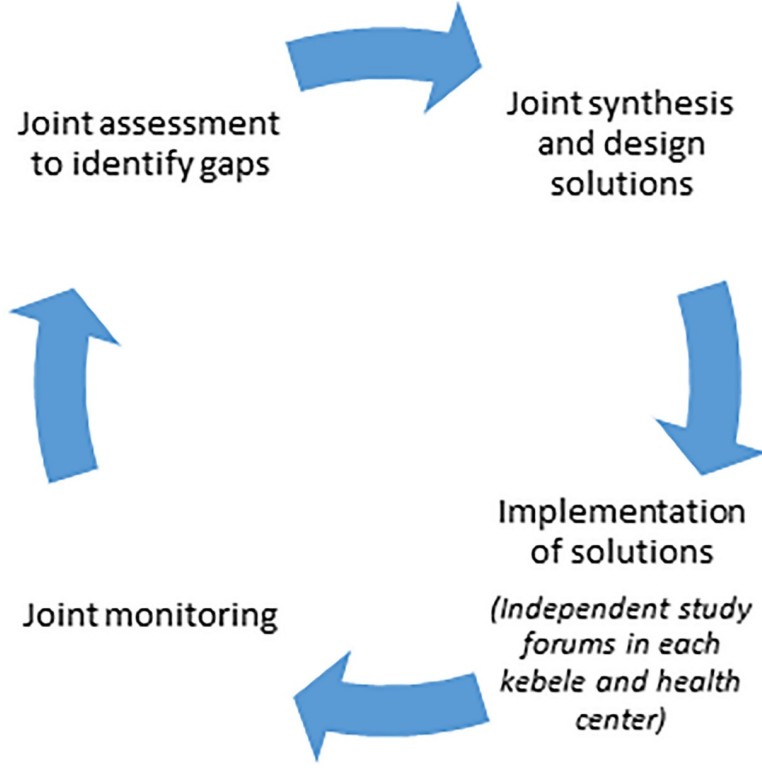

**Fig 1. Participatory community QI cycle.**

people, to help meet HSTP MNH targets. Its strategies include community-based data for decision-making (CBDDM), family conversation, and birth notification [46].

CBDDM is used to identify pregnant women and ensure they receive ANC, intrapartum care, and PNC for themselves and their infants [46]. L10K 2020 also introduced a birth notification strategy to promote postnatal care. Family conversation is a forum conducted at the home of a pregnant woman with her family members and relatives who are encouraged to support her during pregnancy, labor, delivery, and the postpartum period to promote birth preparedness and essential newborn care [47].

**Intervention description.** The project implemented the PC-Solutions strategy in eight of the 115 L10K 2020 Platform woredas (two in each region) between March 2016 and October 2017. This integrated intervention joined communities (including health posts) and facilities (health centers) and built on two previous JSI interventions; Participatory Community Quality Improvement and Early Care-Seeking and Referral Solutions [37, 48, 49]. PC-Solutions emphasized quality of care throughout the primary health care level by including communities, along with providers and managers at the primary health care level and the woreda health office, as a critical constituency.

The PC-Solutions strategy is a four-step QI process (plan-do-study-act), for MNH services provided at PHCUs (Fig 1). In the first step, a joint situational analysis was conducted in the PHCUs using workflow mapping, client exit interviews, document review, and focus group discussions with mothers and WDA members. Following the assessment, a meeting with community members, HEWs, health center staff, woreda health office staff, and referral hospital staff was held to discuss assessment findings, consolidate points, and identify priority problems and their solutions.

Challenges identified during the assessment included delayed ANC booking, suboptimal use of PNC, and poor quality ANC, obstetric care, and PNC. The main problems were: 1) delayed identification of pregnant women and linking them to health center and health post/HEW; 2) inability of health workers to use partograph; and 3) delayed notification of recently delivered women to HEWs for PNC. Community members also mentioned cultural taboos on disclosing pregnancy before a certain amount of time (i.e., people believe that the pregnancy might end in miscarriage if mothers disclosed earlier); lack of awareness (even among WDA members) of the importance of early ANC booking; suboptimal use of PNC; and late notification of births to the HEWs hindering ability to initiate PNC immediately following birth.

Encouraging early ANC booking, increasing use of PNC, and improving quality of ANC, obstetric care, and PNC were PC-Solutions' priorities. Its interventions included early identification and notification system of pregnancy and postpartum mothers; introduction of ANC defaulter tracing mechanism through the WDA members; using mentors and peer learning for onsite partograph training for health care providers; introduction of automated monitoring tools at PHCUs to use data for decision making; and establishing health facility QI teams that included community members. The strategy also included monthly follow up and coaching visits from L10K 2020; monthly QI meetings at health centers and in communities for health post staff, HEWs, and WDA members to review data and progress; quarterly learning sessions; and QI refresher training for facilitators (Table 1).

In the first plan of the PC-Solution strategy, early ANC and PNC, continuity of ANC visits, and partograph use were prioritized and implemented over two years. Intrapartum and newborn care quality were introduced after a reassessment at the end of fiscal year 2017.

Quality improvement teams were formed at the health center and community levels and included health center service providers, HEWs, WDA members, and local administrators. The QI team collated and triangulated administrative data from health centers and posts to inform QI cycle plan and study fora.

The overall QI approach of PC-Solutions included joint "planning and acting" and independent kebele- and the health center level "do and study" fora. The kebele-level do and study cycles were facilitated by health center staff with participation of the kebele-level QI team. Community members identified bottlenecks and solutions and helped implement and monitor the process to improve facility quality and performance.

### Study design

We evaluated the effects of PC-Solutions strategy using mixed-methods research. Four rounds of cross-sectional surveys of all eight health centers were conducted in March 2016, October 2016, April 2017, and October 2017.

A pre-/post-test nonequivalent group study design was nested within the household surveys of women with children 0–11 months conducted in March 2016 and November 2017 to monitor MNH care behaviors and practices in the 115 L10K 2020 intervention areas [50]. To evaluate the effectiveness of the PC-Solution strategy, changes in household MNH care behavior and practices between the two surveys were compared between L10K 2020 Platform areas with and without the PC-Solutions strategy.

Researchers choose a programmatic qualitative research design to answer the qualitative objectives. The interview technique was face-to-face in-depth interviews (IDIs) of WDA members, HEWs, health center directors, health center staff, and L10K 2020 QI specialists from four PHCUs. The qualitative study was conducted in September 2018.

**Table 1. Intervention descriptions.**

| Intervention | Description |
|---|---|
| Continuous QI process | During the implementation phase, internal learning sessions, technical support, and regular performance reviews were conducted at the health center and community levels. |
| | QI teams were established at the health center and at each community. Most of the technical staff from the health center participated in the implementation of the strategy. In each kebele, one person from the health center coordinated community-level implementation. |
| | Introduced automated monitoring tools at PHCUs for data-based decision |
| **Change ideas implemented:** | |
| Early pregnancy identification and birth notification systems | *Used local structures and forums*: places where women meet, like hairdressers in Tigray, religious ceremonies, and monthly meetings with WDA members have been used to identify pregnant women. |
| | *Used local wisdom*: WDAs identified pregnancies for early ANC checks. Among the pregnancy signs used to identify pregnancy were lack of appetite; vomiting; stomach ache; exhaustion; expression of discomfort on face; and headaches. |
| | *Implemented use of pregnancy identification and birth notification cards*: Once the WDAs identified pregnant mothers in their catchment, they gave HEWs pregnancy identification cards to book early ANC. Health center staff sent a birth notification card for facility births to HEWs to initiate PNC as soon as the mother returned home. WDA members sent notification cards to HEWs for home births. |
| | *Introduced defaulter tracing mechanism through WDA members* using pregnancy notification card |
| Provided MNH-related education to WDA and community members | HEWs taught the WDA and community members to encourage mothers to get ANC as early as possible. WDA members then mobilized community through social events such as coffee ceremonies, using peers during marketing and other community events |
| Improved quality of intrapartum and newborn care | Used checklists when providing ANC and PNC |
| | Held onsite partograph training for health care providers using mentors and peer learning |
| | Introduced quality of obstetric and newborn care after reassessment at the end of 2017 fiscal year. |
| Performance reviews | Held monthly QI meetings at health centers and posts to review data and progress and take corrective actions; quarterly learning sessions; and review meetings. |
| Monthly follow-up and coaching visit | Used standard checklist and technical support from L10K 2020 and woreda health office staff at health centers and in communities. Support focused on staff technical competence, the content of care, and quality measurement (record-keeping, data analysis, and use). Performances were measured against stated aims; reasons or challenges that underpinned change ideas were identified; joint discussion held to narrate action points, and way forward directed. |

## Sample size and data collection

All eight health centers were visited for the facility survey. Data were collected through interviews with providers and a review of patient records and service statistics.

For the household surveys, the sample size was powered to detect 10 percentage-points difference between two survey periods for an indicator with alpha error set at 0.05; beta error set at 0.20; and cluster survey design-effect set at 2.0 for the comparison area and 1.0 for the intervention area. The point estimates for an indicator at baseline and follow-up were assumed to

be 45% and 55%, respectively, to yield the largest sample size to detect the desired change. Accordingly, the sample size for the intervention area was determined to be 400 women with children ages 0–11 months; for the comparison area, the sample size was 800 women with children ages 0–11 months.

The household surveys employed a two-stage cluster sampling method stratified by program domain and region. Within each stratum, kebeles were selected as primary sampling units with the probability of selection being proportionate to population size (first stage); at the second stage, the sampling strategy described by Lemeshow and Robinson (1985) [51] was used to select households with target respondents and interview them. To do so, a kebele was sub-divided into three equal segments (sub-kebeles) and four respondents from each segment were interviewed. To identify the first respondent, the interviewers went to the population center of the segment (the point in the segment where the population is about equally distributed on all sides), spun a pen on the ground, and chose the first household in the direction that the pen pointed after it stopped spinning. Consecutive households were visited until the desired sample size was achieved, moving away from the middle of the segment. If the household had women with children 0–11 months and they consented, they were interviewed.

Kebeles visited for data collection during the baseline survey were revisited during the follow-up survey.

A total of 39 kebeles in the intervention area were selected for the study. The strategy estimated that about 80 kebeles from the comparison area would be available to obtain the required sample size. However, as the L10K 2020 innovations were not scaled-up to other woredas as initially planned, the domain of the comparison area included more woredas. Therefore, the number of kebeles representing the comparison area was 148, and the power of the sample was greater than 80% to detect a minimum of 10 percentage points intervention effect. The final sample size of respondents at baseline and follow-up were 2,268 (473 interventions and 1,795 comparison) and 2,244 (468 intervention and 1,776 comparison), respectively.

Data were collected using a structured interview questionnaire (S2 Appendix) designed into an Android mobile application SurveyCTO collect [52].

An interview guide with open-ended questions was used to capture qualitative information from informants (S3 Appendix). We recruited four research assistants, (one per region), who spoke the local language; who have a health background; and who have experiences in qualitative research, to do the interview. We oriented research assistants to maintain neutral so as to not influence the participants' responses. Moreover, we thoroughly discussed with the research assistants on interview techniques to engage participants throughout the interview and to get a truthful and honest answer.

Stratified purposive sampling schemes were used to obtain in-depth information. First, we selected one intervention PHCU from each region to gather detail and contextually relevant data. Then, in each PHCU, staff from the health center and all health posts and selected active WDA members in catchment kebeles were recruited for the study. The research team interviewed the PHCU director, PHCU staff who were actively participating in the implementation of the PC-Solutions strategy, and those who facilitated the community-level QI cycle. We also interviewed the L10K 2020 technical specialists for the PC-Solutions strategy in each region. Through the facilitation of the PHCU director and regional staff, the research team approached HEWs and WDA members and invited them to participate in IDIs at the health post and their home, respectively.

Theoretical sampling technique was used to collect rich information from community health workers until saturation of categories with data is achieved. In-depth interviews of HEWs and WDAs were conducted until saturation of information is reached. All health workers who were actively participating in the implementation of the PC-Solutions strategy and

**Table 2. Outline of participants for qualitative research.**

| Respondent category | Workplace | Number | | | | Total |
|---|---|---|---|---|---|---|
| | | Amhara | Oromia | SNNP | Tigray | |
| L10K 2020 field QI specialists | L10K 2020 regional offices | 1 | 1 | 1 | 1 | 4 |
| Health workers (nurse/midwives/health officers) | Intervention health centers at primary health care level | 3 | 3 | 2 | 4 | 12 |
| Community health workers: | | | | | | |
| HEWs | Frontline grass root level health workers | 5 | 4 | 5 | 4 | 18 |
| WDA members | Community volunteers | 4 | 4 | 4 | 5 | 17 |
| Total | | 13 | 12 | 12 | 14 | 51 |

L10K 2020 QI specialists were included. Accordingly, 51 IDIs were conducted with WDAs, HEWs, health center directors, health center staff and L10K 2020 QI specialists (Table 2). All recruited participants participated, no one refused to participate in the study.

## Measurements

The dependent variables of interest were the household and provider MNH care behaviors and practices that were expected to be affected by the intervention measured by the household survey.

MNH care indicators and facility readiness and performance definitions are in Table 3.

The independent variables that were considered as potential confounders were the individual-, household-, and kebele-level sample characteristics and administrative regions. The individual-level characteristics considered were age, education, marital status, parity, and religion; the household-level characteristics were wealth and distance of the respondents' household from the nearest health facility; and the kebele-level characteristic considered was region.

The wealth index score was constructed for each household with the principal component analysis of the household possessions (electricity, watch, radio, television, mobile phone, telephone, refrigerator, table, chair, bed, electric stove, and kerosene lamp), and household characteristics (type of latrine and water source). The households were ranked according to the wealth score and then divided into five quintiles [53].

## Analysis

Stata 15.1 was used for the statistical analysis conducted for this study [54]. The health facility indicators were presented by survey periods.

The background characteristics of household respondents were compared between study arms at baseline and at follow-up survey periods using Pearson's chi-squared statistics. Similarly, the unadjusted MNH indicators were compared.

To estimate the adjusted intervention effects, the propensity scores were first estimated for each kebele using a logit model that predicted the kebeles in the intervention area at baseline. The covariates of the logit model were kebele averages of individual and household characteristics at baseline, kebele averages of MNH care behavior and practice indicators at baseline, kebele characteristics at baseline, and administrative regions. Covariates that had less than 0.2 p-value in the logit model were dropped using stepwise-backward selection [55, 56]. The final logit model from the stepwise procedure included the following covariates: education, religion, administrative region, first ANC, ANC in the first trimester, complete ANC, ANC experience score, ANC in the first and last trimester, PNC within 48 hours at home and at facility for the mother, and home birth notification. Intervention and comparison kebeles with similar propensity scores at baseline were coded so that they could be identified as similar.

**Table 3. Definition of MNH care and facility readiness and performance indicators.**

| Indicators | Definitions |
|---|---|
| **Readiness and performance of the health centers** | |
| BEmONC signal functions | EmONC is a set of life-saving interventions that treat the major obstetric and newborn causes of morbidity and mortality. To assess the level of care, these functions are classified as basic (BEmONC) or comprehensive (CEmONC). |
| | BEmONC services comprise: 1) administration of parenteral antibiotics to prevent puerperal infection or treat abortion complications; 2) administration of parenteral anticonvulsants for treatment of eclampsia and preeclampsia; 3) administration of parenteral uterotonic drugs for postpartum hemorrhage; 4) manual removal of the placenta; 5) assisted vaginal delivery (vacuum extractions); 6) removal of retained products of conception; and 7) newborn resuscitation. |
| | The following newborn functions were also assessed: 1) antibiotics for preterm or prolonged premature rupture of membrane (PROM) to prevent infection; 2) corticosteroids in preterm labor; 3) kangaroo mother care (KMC) for premature/very small babies; 4) alternative feeding if baby is unable to breastfeed (breast milk expression and cup/spoon-feeding) [12]; and 5) injectable antibiotics for newborn sepsis [12] |
| Readiness to perform signal functions | Health centers' readiness to provide the BEmONC signal functions was defined as the availability of equipment, commodities, and drugs (yes/no response). The specific items linked to each signal function are shown below. |
| Administer parenteral antibiotics | Availability of injectable gentamicin, ampicillin, metronidazole, OR ceftriaxone. |
| Administer uterotonic drugs | Availability of parenteral oxytocin. |
| Administer parenteral anticonvulsants | Availability of magnesium sulfate or diazepam. |
| Manually remove the placenta | Availability of round-the-clock manual removal of placenta services and of at least one BEmONC trained provider. |
| Remove retained products of conception | Availability of manual vacuum aspiration or E&C/D&C set and at least one BEmONC-trained provider. |
| Perform assisted vaginal delivery | Availability of vacuum extractor and at least one BEmONC-trained provider. |
| Perform basic neonatal resuscitation | Availability of Ambu-bag and mask (both small for preterm babies and normal masks) and at least one BEmONC-trained provider. |
| Antibiotics for preterm or prolonged PROM to prevent infection | Availability of oral erythromycin and ampicillin or ceftriaxone. |
| Corticosteroids in preterm labor | Availability of parenteral corticosteroids (betamethasone/dexamethasone). |
| KMC for premature/very small babies | Availability of dedicated space for KMC and trained staff. |
| Alternative feeding if baby is unable to breastfeed (breast milk expression and cup/spoon-feeding) | Availability of utensils for breast milk expression and cup feeding. |
| Injectable antibiotics for newborn sepsis | Availability of injectable ampicillin and gentamicin. |
| Partograph use rate | Of deliveries in the health centers in the past one month, proportion whose labor was monitored using partograph. |
| Uterotonics given for active management of the third stage of labor (AMTSL) | Of deliveries in the health facilities in the past one month, proportion given uterotonics immediately after delivery to prevent postpartum hemorrhage (PPH). |
| **Women's care-seeking behavior** | |
| At least one ANC | % of women who visited a health facility for check-up during her last pregnancy at least once |

(*Continued*)

**Table 3.** (Continued)

| Indicators | Definitions |
|---|---|
| ANC in 1st trimester | % of women who visited a health facility for check-up during the first trimester of last pregnancy |
| ANC 4+ | % of women who went to a health facility for antenatal care at least 4 times during the last pregnancy |
| ANC in 1st & last trimester | % of women who went to a health facility for check-up during her first trimester and during her last trimester of the last pregnancy |
| Skilled birth attendance | % of women who were assisted by a health professional (doctor, nurse, or midwife) during the last childbirth |
| **Providers' service provision behavior** | |
| Complete ANC | % of women who had their blood pressure measured, and blood and urine tested during last pregnancy |
| ANC consultation experience (perceived ANC) | ANC experience is an index constructed using the following survey items: 1) How respectfully were you treated in the health center (or health post)? Would you say very respectfully, respectfully, disrespectfully, or very disrespectfully (a 4-point Likert-type scale); 2) In your opinion, how knowledgeable was the health professionals in the health center (or health post)? Would you say very knowledgeable, knowledgeable, or not knowledgeable (a 3-point Likert-type scale); 3) Overall, how comfortable are you at the health center (or health post)? Would you say very comfortable, comfortable, uncomfortable, or very uncomfortable (a 4-point Likert-type scale); and 4) how responsive was the health center (or health post) to your needs? Would you say very responsive, responsive, unresponsive, or very unresponsive (a 4-point Likert-type scale)? To account for the difference in the number of points in the Likert-type responses of the items, the mean of each of the items was standardized and then aggregated to obtain the ANC experience index. We reversed the index so that higher score indicated relatively better ANC experience. The index score was recalibrated to range between 0 and 10. |
| | Cronbach's alpha (reliability coefficient) for the 4 items was 0.87. However, the distribution is found to be skewed. As such, we categorized into 2 groups; those who have the maximum score and others. |
| ANC counseling | ANC counseling is an index constructed using binary response items measuring whether a woman received ANC counseling for 1) breastfeeding; 2) postpartum family planning; 3) HIV; 4) maternal nutrition; 5) danger signs of pregnancy; 6) birth preparedness and complication readiness; and 7) newborn care. The yes responses were coded 0 and no coded 1, and the 7 items aggregated to construct the index. The index score was recalibrated to range between 0 and 10, with a higher score indicating better ANC counseling. |
| | Cronbach's alpha for the 7 items was 0.86. Then, we categorized into 2 groups; those with the maximum score and others. |
| Perceived knowledge of providers | This is based on delivery experience at the health facility. Women were asked, "In your opinion, how knowledgeable are the health professionals in the health center? Would you say very knowledgeable, knowledgeable, or not knowledgeable?" |
| | "Very knowledgeable and knowledgeable" are categorized as knowledgeable. |

(*Continued*)

**Table 3.** (Continued)

| Indicators | Definitions |
|---|---|
| Satisfaction with delivery care | Women were asked, "If a close friend of yours were pregnant, would you recommend that she deliver at the same facility where you did, at another health facility, or would you recommend that she not deliver at any health facility?" If they recommended going to the same facility, it is categorized as "satisfied;" otherwise categorized as "not satisfied." |
| Disrespect and abuse | Disrespect and abuse were defined if a woman experienced any of the following categories of disrespect and abuse during childbirth in a facility: 1) physical abuse; 2) treatment without permission; 3) violate privacy; 4) violate confidentiality; 5) verbal abuse; and 6) left unattended. |
| PNC within 48 hours for the mother (home) | % of women who received postpartum care at their home within 48 hours of last childbirth |
| PNC within 48 hours for the baby (home) | % of women who received newborn care at their home within 48 hours of last childbirth |
| PNC within 48 hours for the mother (both at home and facility) | % of women who received postpartum care at the health facility or at their home within 48 hours of last childbirth |
| PNC within 48 hours for the baby (at home and facility) | % of women who received newborn care at the health facility or at their home within 48 hours of last childbirth |
| Stayed in facility for 24 hours or more | % of women who stayed for 24 hours or more in the facility after the delivery |
| Birth notification (home birth) | % of women who delivered at home and took measures to inform the HEW about childbirth immediately after delivery |
| Birth notification (institutional birth) | % of women who delivered at facility and took measures to inform the HEW about childbirth immediately after delivery |
| Used motor vehicle transport | % of women among those who gave birth at facility and used motor vehicle transport to get there |

To assess the adequacy of the matching, t-tests were performed to ensure that the covariates of the final logit model were not statistically significantly ($p > 0.1$) between the intervention and the control kebeles, after accounting for the matched kebeles.

Finally, intervention effects (difference-in-difference) was estimated from kebele-level random effect models predicting the outcome of interest with indicator variables for study arm, survey period, the interaction term between study arm and survey period, and for the kebeles that matched between the intervention and comparison areas (dummy variables) as the predictors. Stata's 'margins' command was used to obtain adjusted estimates of the outcomes of interest according to study arm and survey period and the difference-in-difference (DiD) (i.e. intervention effects).

For the qualitative component of the study, audio records from IDIs were transcribed verbatim. The data were analyzed thematically. The transcript texts were manually coded. Then, themes were derived from the data coded. The codes, categories and the concepts emerged from an interview group were verified by linking the emerging categories with the data received from another group of informants to improve the trustworthiness of the qualitative data analysis. These categories were also linked to quotes from the research informants to ensure the reliability of the study. During reporting, participant quotations are presented to illustrate the themes/findings.

### Ethics approval and consent to participate

Ethical clearances for the surveys were obtained from the ethical review boards of Amhara, Oromia, SNNP, and Tigray Regional Health Bureaus, and JSI. All participants were informed about the purpose of the study; benefits and hazards of the study were explained to all study participants, and each participant was notified of his/her right to opt out when responding to questions. Verbal consent was sought and documented before conducting any interviews. If the respondent was younger than 18 years old, consent was sought from her husband or guardian. Because the majority of the respondents were not expected to be able to read or write; written consent was not sought. If the respondent agreed to be interviewed after listening to the consent statement, the interviewer marked the questionnaire as consent given below the consent statement and signed below that. The interviewer continued with the interview only after receiving and documenting consent. The survey protocol submitted to the ethical review committee included the study questionnaire with the statement that described the consent-obtaining procedure. Moreover, the information obtained from the research participants was kept private (codes were used during reporting of the IDI quotes).

## Results

The evaluation results are presented as follows; first, the MNH indicators and BEmONC functions from the facility surveys; followed by, the quasi-experimental study findings; and finally, the qualitative study findings.

### The MNH indicators and the BEmONC functions from the facility survey

**Performance of BEmONC signal functions.** Based on the definition of functional BEmONC facility [57], about half of the health centers were ready to provide the seven functions, but only 13% were doing so in the last three months. Likewise, while 69% of them were ready to provide six BEmONC functions, only about 13% of health centers were doing so, with no significant change over time (Table 4).

A significant number of health centers lacked provision of parenteral anticonvulsants and manual removal of placenta. Moreover, most lacked readiness and/or provision of most newborn signal functions including KMC, alternate feeding for non-breastfeeding babies, antibiotics

**Table 4. Health center readiness to provide BEmONC signal functions (%).**

| Signal functions | March 2016 | October 2016 | April 2017 | October 2017 |
|---|---|---|---|---|
| Parental antibiotics | 100.0 | 100.0 | 100.0 | 100.0 |
| Parenteral uterotonic | 100.0 | 100.0 | 100.0 | 100.0 |
| Parenteral diazepam/MgSO4 | 87.5 | 100.0 | 87.5 | 100.0 |
| Removal of retained products | 87.5 | 87.5 | 87.5 | 75.0 |
| Manual removal of placenta | 100.0 | 100.0 | - | 100.0 |
| Assisted vaginal birth | 100.0 | 100.0 | 100.0 | 62.5 |
| Newborn resuscitation | 87.5 | 100.0 | 87.5 | 75.0 |
| Antibiotics for pPRoM | 87.5 | 62.5 | 75.0 | 50.0 |
| KMC | 0.0 | 0.0 | 25.0 | 0.0 |
| Corticosteroid | 25.0 | 37.5 | 87.5 | 37.5 |
| Newborn sepsis antibiotics | 75.0 | 62.5 | 62.5 | 50.0 |
| Alternate feeding for not breastfeeding | 12.5 | 12.5 | 12.5 | 25.0 |
| Readiness to perform the 7 signal functions | 62.5 | 87.5 | 62.5 | 50.0 |
| Readiness to perform the 6 signal functions | 62.5 | 87.5 | 62.5 | 62.5 |

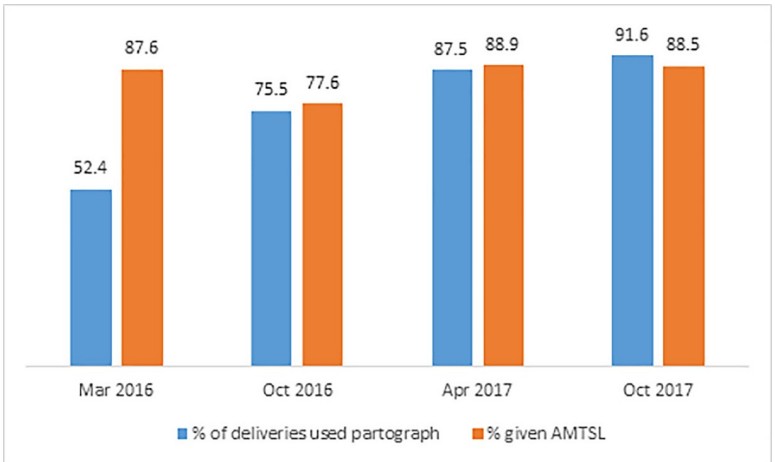

**Fig 2. Partograph use and administration of prophylactic uterotonics.**

for preterm premature rupture of membrane, corticosteroids for preterm labor, and antibiotics for sepsis.

**Coverage and quality of the facility-level MNH interventions.** Integrated maternal and newborn health care of mothers who gave birth in the last month at the surveyed health centers were retrieved for review. In each period, about 36 delivery records (mothers' chart) per health center (a total of 1,014) were reviewed to assess the active management of third-stage labor and use of partograph. Seventy-six percent (774) of them used partograph and 86% (870) mothers were given prophylactic uterotonics for AMTSL to prevent PPH. Fig 2 shows that trends in partograph use and administration of prophylactic uterotonics increased over the survey period.

During each survey period, data on number of deliveries, delivery outcomes, obstetric complications managed, low-birth-weight babies, and newborn sepsis and asphyxia were collected retrospectively for six months from surveyed health centers to understand the service use. The syphilis-testing rate increased from 26% in March 2016 to 38% in October 2017. The facility delivery rate increased from 57% in March 2016 to 64% in October 2017. The fulfilled need for BEmONC stalled at about 18% on average. Also, the percent of expected possible serious bacterial infection (PSBI) cases managed at health centers declined over the survey period.

The proportion of asphyxiated newborns managed with Ambu-bag and mask increased from 29% to 93% during the last two years. The proportion of preterm/low-birth-weight babies initiated KMC increased from 13% to 92% in the same period.

## Sample characteristics of the household survey respondents

As depicted in Table 5, respondents were significantly different in their education distribution between the study arms. During both survey periods, intervention arm respondents were more likely to have higher education. However, during both survey periods, respondents of the two study arms were similar in terms of age, administrative region, and household wealth.

## Unadjusted estimates of MNH indicators

At baseline, the intervention arm had significantly higher coverage of MNH indicators including first ANC, ANC 4, complete ANC, ANC experiences mean score, birth notification, and use of motor vehicle transport than the comparison arm. On the other hand, the intervention

**Table 5. Sample characteristics, by study arm and survey period.**

| Characteristics | Baseline | | | Follow-up | | |
|---|---|---|---|---|---|---|
| | Comparison | Intervention | p-value | Comparison | Intervention | p-value |
| Age group | | | | | | |
| 15–19 | 138 (7.7) | 27 (5.7) | 0.332 | 104 (5.9) | 32 (6.8) | 0.723 |
| 20–34 | 1,384 (77.1) | 374 (79.1) | | 1,374 (77.4) | 357 (76.3) | |
| 35–49 | 273 (15.2) | 72 (15.2) | | 298 (16.8) | 79 (16.9) | |
| Education | | | | | | |
| No education | 1,063 (59.2) | 235 (49.7) | **0.001** | 989 (55.7) | 226 (48.3) | **0.016** |
| Primary | 395 (22.0) | 130 (27.5) | | 397 (22.4) | 124 (26.5) | |
| Higher | 337 (18.8) | 108 (22.8) | | 390 (22.0) | 118 (25.2) | |
| Distance to a health facility | | | | | | |
| < 30 minutes | 752 (41.9) | 185 (39.1) | 0.537 | 839 (47.2) | 215 (45.9) | 0.859 |
| 30 minutes to < 1hr | 601 (33.5) | 168 (35.5) | | 537 (30.2) | 143 (30.6) | |
| 1 + hrs. | 442 (24.6) | 120 (25.4) | | 400 (22.5) | 110 (23.5) | |
| Wealth quintile | | | | | | |
| Most poor | 412 (23.0) | 97 (20.5) | 0.131 | 356 (20.1) | 88 (18.8) | 0.812 |
| More poor | 366 (20.4) | 82 (17.3) | | 349 (19.7) | 88 (18.8) | |
| Poor | 337 (18.8) | 109 (23.0) | | 347 (19.5) | 91 (19.4) | |
| Less poor | 336 (18.7) | 85 (18.0) | | 352 (19.8) | 91 (19.4) | |
| Least poor | 344 (19.2) | 100 (21.1) | | 372 (21.0) | 110 (23.5) | |
| Region | | | | | | |
| Tigray | 408 (22.7) | 108 (22.8) | 0.990 | 396 (22.3) | 108 (23.1) | 0.982 |
| Amhara | 469 (26.1) | 120 (24.4) | | 468 (26.4) | 120 (25.6) | |
| Oromia | 456 (25.4) | 122 (25.8) | | 456 (25.7) | 120 (25.6) | |
| SNNP | 462 (25.7) | 123 (23.0) | | 456 (25.7) | 120 (25.6) | |
| No. of women | 1,795 | 473 | | 1,776 | 468 | |

arm had significantly lower coverage of ANC counseling mean score and perceived knowledge of providers. Likewise, at follow-up period, the intervention arm had higher coverage of MNH indicators except for ANC 4+, complete ANC, ANC counseling score, women's satisfaction with delivery care, disrespect and abuse, and use of motor vehicle transport (Table 6).

## Intervention effects in MNH care practices

The intervention effects were adjusted for the baseline differences in the kebele-level MNH indicators and socio-demographic characteristics between the study arms (Tables 6 and 7, respectively), using the propensity score matching (PSM) technique. The assessment of the balance of the sample characteristics and MNH indicators at baseline was found to be adequate. Between the baseline and follow-up surveys, ANC in the first trimester increased by 5.6 percentage points (from 26.5% to 32.1%) in the comparison area, which was statistically significant ($p < 0.05$). ANC in the first trimester increased by 13.3 percentage points (from 25.6% at baseline to 38.9% at follow-up) in the intervention area, which was also statistically significant ($p < .01$). The increase in the coverage of that indicator was 7.6 percentage points higher in the intervention area than in the comparison area, which is attributable to the PC-Solutions strategy. In other words, the intervention effect (or the DID) was 7.6 percentage points. But the two-sided p-value of the statistical significance of the intervention effect was just above 0.05 (0.051), indicating it was not statistically significant. Nonetheless, the p-value for testing one-sided hypothesis (i.e., the increase in the coverage of the indicator in the intervention arm was higher than that in the comparison arm) was statistically significant ($p = 0.026$).

**Table 6. Maternal and newborn health care practices between study arms and survey periods.**

| MNH indicators | Baseline | | | Follow-up | | |
|---|---|---|---|---|---|---|
| | Comparison % (N) (95% CI) | Intervention % (N) (95% CI) | p-value | Comparison % (N) (95% CI) | Intervention % (N) (95% CI) | p-value |
| **Women's care-seeking behavior** | | | | | | |
| % at least one ANC | 85.4 (1,792) (83.2, 87.7) | 87.8 (471) (86.0,89.6) | < **0.001** | 92.6 (1,774) (91.0, 94.2) | 468 (96.8) (94.9, 98.8) | < **0.001** |
| % ANC in 1st trimester | 22.8 (1,758) (20.6, 24.9) | 23.8 (463) (21.8, 25.8) | 0.100 | 29.2 (1,729) (26.4, 32.0) | 458 (36.7) (30.6, 42.7) | **0.013** |
| ANC 4+ | 48.7 (1,758) (45.6, 51.8) | 50.4 (463) (47.5, 53.3) | **0.013** | 55.3 (1,729) (51.9, 58.7) | 58.8 (458) (52.8, 64.9) | 0.221 |
| % ANC in 1st & last trimester | 21.1 (1,758) (19.1, 23.2) | 22.2 (463) (20.3, 24.2) | 0.070 | 27.6 (1,729) (24.9, 30.4) | 32.3 (458) (28.4, 40.2) | **0.024** |
| Skilled delivery | 65.1 (1,765) (61.3, 68.8) | 66.3 (470) (62.7, 69.8) | 0.053 | 70.9 (1,761) (67.2, 74.6) | 77.7 (460) (72.7, 82.8) | **0.003** |
| **Providers' service provision behavior** | | | | | | |
| % complete ANC | 47.2 (1,792) (43.7, 50.6) | 50.8 (471) (47.5, 54.1) | < **0.001** | 59.4 (1,774) (55.8, 63.0) | 63.3 (468) (57.4, 69.2) | 0.150 |
| ANC consultation experiences | 32.5(1,496) (29.8, 35.1) | 30.0 (422) (27.7, 32.3) | **-0.001** | 27.6 (1,622) (24.8, 30.3) | 34.0 (456) (28.1, 39.9) | **0.026** |
| ANC counseling | 39.8 (1,148) (36.5, 43.2) | 42.1 (312) (39.0, 45.3) | **0.010** | 47.1 (1,360) (43.3, 50.8) | 48.0(401) (741.1, 54.8) | 0.788 |
| Perceived knowledge of providers | 49.3 (1,141) (46.2, 52.4) | 46.0 (322) (43.2, 48.7) | -< **0.001** | 42.1 (1,211) (38.6, 45.5) | 50.0 (358) (42.7, 56.5) | **0.030** |
| Satisfaction with delivery care | 97.3 (1,141) (96.5, 98.1) | 97.8 (322) (97.2, 98.3) | 0.098 | 98.5 (1,211) (97.9, 99.2) | 99.2 (358) (97.9, 100.0) | 0.343 |
| Disrespect and abuse | 10.1 (1,086) (8.3, 11.8) | 9.4 (315) (8.0, 10.7) | -0.208 | 8.4 (1,188) (6.7, 10.1) | 8.8 (352) (3.9, 13.8) | 0.867 |
| PNC in 48 hours of the mother (both home and facility) | 27.7 (1,795) (25.4, 30.1) | 28.1 (473) (25.9, 30.3) | 0.601 | 34.3 (1,776) (31.3, 37.2) | 50.4 (468) (44.1, 56.8) | < **0.001** |
| PNC in 48 hours of the baby (both home and facility) | 26.1 (1,795) (23.7, 28.5) | 26.3 (473) (24.1, 28.5) | 0.758 | 32.3 (1,776) (29.3, 35.2) | 49.4 (468) (42.8, 55.9) | < **0.001** |
| Home PNC in 48 hours (mother) | 6.8 (1,795) (5.7, 7.9) | 7.1 (473) (6.1, 8.1) | 0.386 | 10.2 (1,776) (8.4, 12.0) | 15.2 (468) (10.2, 20.1) | **0.047** |
| Home PNC in 48 hours (baby) | 4.6 (1,795) (3.7, 5.5) | 4.7 (473) (3.9, 5.5) | 0.739 | 7.0 (1,776) (5.4, 8.6) | 12.7 (468) (7.4, 18.1) | **0.034** |
| Stayed in facility for 24 hours or more | 22.9 (1,795) (20.6, 25.2) | 23.1 (473) (21.0, 25.2) | 0.799 | 27.9 (1,776) (25.1, 30.8) | 42.6 (468) (35.9, 49.3) | < **0.001** |
| Birth notification (home birth) | 47.2 (1,255) (44.0, 50.4) | 49.3 (358) (46.3, 52.3) | **0.012** | 58.4 (1,341) (54.9, 61.9) | 68.9 (406) (63.1, 74.6) | < **0.001** |
| Birth notification (institutional birth) | 42.0 (1,156) (38.8, 45.3) | 44.0 (326) (41.0, 47.1) | **0.025** | 53.5 (1,219) (49.8, 57.2) | 65.6 (366) (59.3, 71.9) | < **0.001** |
| Used motor vehicle transport | 53.7 (1,154) (49.2, 58.2) | 55.8 (324) (51.4, 60.1) | **0.009** | 59.5 (1,218) (54.9, 64.1) | 58.1 (358) (50.9, 65.3) | 0.672 |

We observed statistically significant intervention effects on skilled delivery; PNC (at home and health facility) of the mother in 48 hours; PNC (at home and health facility) of the new-born in 48 hours; and stayed in facility for 24 hours or more, were 7.9% (1.8–13.9); 15.3% (7.4–23.2); 17.0% (9.1–24.8); and 13.5% (5.7–21.4), respectively.

**Table 7. Propensity score matched DiD treatment effect estimations of MNH care practices, by survey period and study arm.**

| MNH indicators | Intervention | | | | Comparison | | | | Difference-in-difference | |
|---|---|---|---|---|---|---|---|---|---|---|
| | Baseline | Follow-up | Diff (C-I) | p-value | Baseline | Follow-up | Diff (C-I) | p-value | DiD | p-value |
| **Women's care-seeking behavior** | | | | | | | | | | |
| % at least one ANC | 89.6 | 97.5 | 7.8 (5.5–10.2) | <**0.001** | 89.2 | 94.1 | 4.9 (2.5–7.2) | <**0.001** | 3.0 (-0.0–6.0) | 0.053 |
| % ANC in 1st trimester | 25.6 | 38.9 | 13.3 (7.1–19.4) | <**0.001** | 26.5 | 32.1 | 5.6 (1.1–10.2) | **0.015** | 7.6 (-0.0–15.3) | 0.051 |
| ANC 4+ | 52.6 | 60.9 | 8.3 (2.5–14.1) | **0.005** | 51.7 | 59.7 | 8.1 (3.5–12.6) | <**0.001** | 0.2 (-7.2–7.6) | 0.954 |
| % ANC in 1st & last trimester | 24.4 | 37.0 | 12.6 (6.5–18.7) | <**0.001** | 25.1 | 30.8 | 5.6 (1.1–10.2) | **0.014** | 6.9 (-0.7–14.6) | 0.075 |
| Skilled delivery | 68.2 | 79.5 | 11.2 (6.6–15.9) | <**0.001** | 69.3 | 72.7 | 3.4 (-0.5–7.2) | 0.089 | 7.9 (1.8–13.9) | **0.011** |
| **Providers' service provision behavior** | | | | | | | | | | |
| % complete ANC | 54.7 | 66.9 | 12.2 (6.8–17.6) | <**0.001** | 53.3 | 65.1 | 11.8 (7.5–16.1) | <**0.001** | 0.4 (-0.7–7.3) | 0.914 |
| ANC consultation experiences | 29.2 | 33.2 | 4.0 (-0.2–10.0) | 0.199 | 30.6 | 29.0 | -1.6 (-6.9–3.6) | 0.543 | 5.6 (-2.4–13.6) | 0.171 |
| ANC counseling | 41.6 | 47.6 | 6.0 (-1.0–13.0) | **0.093** | 38.3 | 54.2 | 15.9 (9.8–21.9) | <**0.001** | -9.8 (-19.2–0.5) | **0.039** |
| Perceived knowledge of providers | 42.5 | 46.3 | 3.7 (-3.5–11.0) | 0.314 | 44.3 | 40.9 | -3.4 (-9.2–2.4) | 0.254 | 7.1 (-2.2–16.4) | 0.134 |
| Satisfaction with delivery care | 97.6 | 99.0 | 1.5 (-0.4–3.3) | 0.114 | 97.4 | 98.4 | 1.0 (-0.6–2.5) | 0.220 | 0.5 (-1.8–2.8) | 0.678 |
| Disrespect and abuse | 8.3 | 8.1 | -0.2 (-5.0–4.6) | 0.929 | 9.2 | 6.8 | -2.4 (-5.2–0.5) | 0.106 | 2.1 (-3.5–7.8) | 0.453 |
| PNC in 48 hours of the mother (both home and facility) | 26.6 | 48.8 | 22.2 (15.9–28.6) | <**0.001** | 28.5 | 35.5 | 7.0 (2.4–11.5) | **0.003** | 15.3 (7.4–23.2) | <**0.001** |
| PNC in 48 hours of the baby (both home and facility) | 24.9 | 47.7 | 22.8 (16.4–29.2) | <**0.001** | 27.1 | 32.9 | 5.8 (1.4–10.3) | **0.010** | 17.0 (9.1–24.8) | <**0.001** |
| Home PNC in 48 hours (mother) | 6.4 | 13.8 | 7.4 (2.9–11.9) | **0.001** | 7.1 | 10.6 | 3.5 (-0.0–7.0) | 0.050 | 3.9 (-1.8–9.6) | 0.177 |
| Home PNC in 48 hours (baby) | 4.2 | 11.6 | 7.5 | **0.002** | 5.0 | 7.4 | 2.4 (-0.7–5.5) | 0.134 | 5.1 (-0.7–10.8) | 0.084 |
| Stayed in facility for 24 hours or more | 22.3 | 41.5 | 19.2 (12.8–25.6) | <**0.001** | 23.8 | 29.5 | 5.6 (1.3–10.0) | **0.010** | 13.5 (5.7–21.4) | **0.001** |
| Birth notification (home birth) | 49.2 | 68.8 | 19.6 (13.4–25.8) | <**0.001** | 48.8 | 63.5 | 14.7 (9.5–19.8) | <**0.001** | 4.9 (-3.3–13.2) | 0.240 |
| Birth notification (institutional birth) | 43.8 | 65.4 | 21.6 (14.9–28.3) | <**0.001** | 43.7 | 59.0 | 15.3 (9.8–20.8) | <**0.001** | 6.3 (-2.6–15.2) | 0.163 |
| Used motor vehicle transport | 58.7 | 61.0 | 2.2 (-4.2–8.6) | 0.493 | 56.9 | 64.3 | 7.5 (2.8–12.2) | **0.002** | -5.2 (-13.2–2.7) | 0.194 |

However, the DiDs were not statistically significant (*p*>.05) for first ANC; complete ANC; four and more ANC visits; perceived knowledge of providers; women's satisfaction with delivery care score; birth notification; motor vehicle use for transport; and experience of any form of disrespect and abuse during childbirth.

## Perceived effect of the intervention

The following perceived effects of the strategy were identified: 1) increased service use and improved quality of care; 2) enhanced knowledge and skill of health workers and provision of standardized care; 3) enhanced community involvement; 4) strengthened linkages between

communities and the formal health care system; and 5) helped to measure and evaluate quality.

**Improved service use and quality of care.** WDA informants mentioned that the project brought tremendous changes in their village as a result of implementing this strategy. As WDA in SNNP said, "Changes are untold compared to the past."

According to the informants, this strategy increased care-seeking behavior of pregnant mothers who participated in the project. The community realized how important beginning first ANC at three months for avoiding pregnancy-related risks, and women began disclosing their pregnancy to network/WDA members and getting ANC in the first trimester. According to the accounts of the interviews, another result of the strategy was that dropout rates across the continuum of MCH care decreased.

**Enhanced the knowledge and skills of health workers.** Informants perceived enhanced health worker capacity in and community engagement in MNH services. The project's participatory design and continuous learning process heightened WDA members' awareness of the quality of MNH services. This knowledge was transferred to WDA members to mothers subsequently.

"The good legacy of the strategy is it showed us what components of MNH services we should provide mothers and newborns." -Health center informant in Amhara.

Participants observed that training and technical assistance filled skill gaps and standardized practice among health workers.

**Strengthened linkages between communities and the formal health care system.** Interview participants also reflected that the PC-Solution strategy helped to enhance linkages between HEWs and WDAs. It also strengthened the linkage between health centers and posts and connected communities to the health system. Respondents said that a strong link has created among WDAs, HEWs, and community. "As positive consequences . . .we established a strong relationship among community, WDAs, HEWs, and health center staff."–Health center informant, SNNP.

**Introduced quality of care indicators with the routine monitoring and evaluation systems of the PHCUs.** Respondents said that they used to evaluate coverage of MNH services; since PC-Solutions, they are measuring and evaluating MNH services in the view of quality. Participants mentioned that they learned how to conduct formative assessment, design solutions, and how to measure performance.

"From PC-Solution, I learned how to review my work with evidence and generate change ideas."–Health center informant, Amhara.

## Facilitators of and barriers to implementation

**Facilitators.** Participants said that full stakeholder participation in all stages of the project, strong coordination, robust support, continuous performance review, and staff commitment facilitated the implementation of the PC-Solutions strategy.

Another key to the strategy's success was high community engagement in QI planning, implementation, and monitoring. Informants mentioned that having shared responsibilities at all levels of the woreda health system and a detailed micro-plan, indicating who is responsible for what and when was also helpful. They also indicated that the development of coordinated activities, especially communication between the community, WDAs, HEWs and health center staff was another success factor.

"The project brought a new idea. It was participatory in that everyone who was supposed to be stakeholder was participating in the project and focused on continuous assessment of the problems and identifying potential solutions, plan accordingly and continues like this to get better results every time the team meets", health center participant in Tigray.

Early PNC adoption was attributed to HEWs, who visited newly delivered mothers at home traveled long distances and often difficult roads and terrain, and to health center staff, who encouraged mothers to stay at facilities least 24 hours post-delivery for early PNC.

**Barriers.** The implementation of this strategy was accompanied by several challenges, including staff turnover at the health centers, the workload of the health workers and HEWs, competing priorities of the health service providers and the WDAs, and magnesium sulfate (MgSO4) and vacuum extractor shortages.

High staff turnover compromised quality improvement because it took 2–3 months to train people in QI. Frequent community leadership change also hindered QI committee implementation. As HEW in SNNP said, "The kebele was not stable for the last two years; leadership was constantly changing. Due to this, QI committee was not working regularly and I can say it wasn't functional. Workload, coupled with lack of HEW, hindered the QI project; I am the only HEW in the kebele and have remote villages that are difficult to reach."

High client caseloads at health center thwarted PC-Solutions implementation. It also overloaded birth attendants, who were unable to attend women in delivery adequately and which caused mothers to give low delivery satisfaction scores. Unmanageable workloads lead staff to provide contact-focused rather than content-focused care, which led to insignificant intervention effects for specific ANC indicators. Respondents reported poor counseling and care quality, which likely contributed to the low use of ANC services. As study participants noted, if ANC counseling is not well organized and pregnant women are not convinced to visit the health center again, it is unlikely that they will come for subsequent appointments.

Due to competing priorities such as campaigns, QI events, in particular meetings, were held irregularly. And when staff were working on other assignments, MNH services were compromised.

Shortages of inputs such as MgSO4 and vacuum extractor sets, which were not available at the market, also contributed to poor performance of BEMONC signal functions at health centers.

## Discussion

This evaluation demonstrates that participatory QI improved use of MNH care services including early ANC care-seeking, skilled delivery, and provision of home- and facility-based PNC for both the mother and the newborn. Interview respondents also perceived that the PC-Solution strategy resulted in a number of changes in the use and quality of MNH services. Participants said that links between communities and health systems and between the health center, health post, and community improved. Communication between WDA and community members improved as WDA member knowledge did.

These findings are in line with a systematic review of participatory learning and action cycles with women's groups reporting improved clean home-delivery practices and uptake of any ANC [35], and with other studies on participatory community QI approaches that reported improved use of maternal health services [33, 37].

In this study, skilled delivery coverage was higher in the intervention than in the control areas. Mixed findings were reported from women's group trials. It is in line with a previous quasi-experimental study in Ethiopia that engaged communities in identifying barriers to access and quality of services and reported an 11 percentage-point increase in average treatment effect in institutional deliveries [37]. But it is inconsistent with other studies elsewhere. For instance, studies from India, Bangladesh, and Malawi reported no effect on increasing health facility deliveries [36, 38, 39, 58–60]. And a systematic review of community-based

interventions packages and a quasi-experimental study in Tanzania and Uganda also did not show significant improvement in skilled attendance at birth [61, 62].

Unlike other studies conducted in Ethiopia, Bangladesh, and India [37–39], this study indicated that community participation had a positive effect on PNC coverage. This might be due to improving HEW home visits regularity, which is supported by previous studies showing that community participation improved the accountability of health care providers [33, 63], and improved birth notification systems and practices of keeping women at facilities for at least 24 hours after delivery for PNC.

Contrary to respondents' opinion that early care-seeking behavior of pregnant mothers improved following the intervention, early ANC booking, and quality of maternal services did not significantly change over time. Complete ANC (ANC 4 and more visits), ANC counseling score, women's satisfaction with delivery care, and experience of any form of disrespect and abuse during childbirth did not show significant improvement in this study. This is in line with a cluster-randomized controlled evaluation of a community participation intervention in Malawi that reported no significant difference in perceived quality of delivery care and sufficient ANC [33]. Although disrespect and abuse during childbirth were not prevalent in this study, the reliability and validity of disrespect and abuse measurements are not well known [64]. As mentioned by respondents, a possible reason for non-significant intervention effects observed for specific ANC indicators is the practice of contact-focused rather than content-focused care. The other possible reason for no improvement in complete ANC, ANC 4 and more visits, and women's satisfaction with delivery care is the short intervention period. These thematic areas were introduced later in 2017 and only had about one year of implementation, so these outcomes took place before the intervention strategies were matured. Moreover, the quality of care measure to events spread over the past 12 months preceding the follow-up survey. As such, many respondents in the intervention area were not exposed to aspects that were introduced during the second part of 2017.

About two-thirds of the health centers were ready to provide the seven BEmONC functions, but only 13% of these did so in the past three months preceding the survey. Though this is much higher than the recent national EmONC survey findings [15], a significant number of health centers did not provide parenteral anticonvulsants or remove placenta manually. Moreover, most health centers lacked readiness and/or provision of most newborn signal functions including KMC, alternate feeding for not breastfeeding babies, antibiotics for preterm premature rupture of membrane, corticosteroids for preterm labor, and antibiotics for sepsis. As such, availability of critical MNH equipment and drugs needs to be improved and facilities need to upgrade their provision of EmONC functions.

Percent of deliveries used partograph and provision of prophylactic uterotonics for AMTSL to prevent PPH increased over the survey periods. The percent of health centers using paragraph was higher than the national EmONC survey [15]. Moreover, access to and use of most maternal and newborn critical interventions, including syphilis testing, facility delivery, asphyxiated newborns managed with Ambu-bag and mask, and preterm/low birth weight babies initiated on KMC improved over time. However, the percent of expected PSBI cases managed at health centers declined over the survey periods. This could be because HEWs began managing PSBI cases through a community-based newborn care program.

This study examined the complex participatory community QI interventions in maternal and newborn health by interviewing those who were involved in the design and implementation of PC-Solutions strategy. Researchers explored the extent of community engagement in the health system and motivating and demotivating factors for sustained engagement in the health system. As such, the findings should be relevant to similar QI projects and stakeholders who intend to scale similar interventions.

The support system, which means facilitating review forums, conducting supervisions and mentoring from partners and/or woreda health office, was critical to the implementation of this participatory intervention. Woreda-level joint reviews were used to evaluate PHCU activities and identify service and knowledge gaps, and launch interventions between communities. They also motivated and capacitate stakeholder knowledge for further engagement and keeping their momentum. Continuous refresher trainings and supportive supervision visits to monitor and coach were helped deliver expected outcomes. Regular performance review and continual support help ensure provision of high-quality health care services.

We did multiple analyses for multiple different endpoint outcomes. Adjusting p-values is often recommended when conducting multiple hypotheses tests 'simultaneously' [65–68]. However, it is not always necessary [69–71]. As such, we did not adjust the p-value in our case as it would have increased the probability of making type II error, i.e., concluding that the intervention was not effective when it is true [69, 70, 72, 73].

Although combining PSM with DiD methods can help resolve the problem of time-invariant unmeasured confounders, the presence of time-varying unobserved confounders would bias the observed treatment effects. For instance, other programs or other developmental inputs (road, health facility construction, etc.) that influence MNH could be kebele-level time-varying or time-invariant confounders. If the government or any other development agency were implementing QI interventions for MNH services in the PC-Solutions areas, then the intervention effects estimated would be biased. However, there is no compelling reason to believe that the government or other development partners were systematically providing inputs in the intervention areas but not the comparison areas. The other major confounder is the Hawthorne effect because the intervention area was not masked. The providers knew that they were under study, which may have led them to perform better than they normally do. Moreover, study results may have been subjected to recall and social desirability biases because the survey used a 12-month recall of self-reported behavior.

We analyzed and presented the consistency between the findings with the quantitative study and the existing body of knowledge. However, one limitation could be that feedback from participants on the findings was not sought.

## Conclusions

The PC-Solutions strategy suggests community engagement in the design and implementation of QI would improve MNH outcomes. Engaging communities in the design of the intervention would yield local solutions to local problems. As such, scale-up of QI initiatives would benefit from the engagement of all relevant local stakeholders throughout the design and implementation. Using community wisdom to implement interventions increases sustainability. Moreover, a strong support system was critical to the implementation of this participatory intervention. Further research is needed in this area before concluding that disrespect and abuse was low.

## Supporting information

**S1 Appendix. Survey dataset.** This is survey data with variables and their values we used for the analysis.
(XLS)

**S2 Appendix. Survey questionnaire.** Survey questionnaire we used to collect information from study participants. The first sheet contains variable definitions (data dictionary) in English and other local languages (Amharic, Oromiffa, and Tigregna), and the second sheet

contains variable answer choices.
(XLSX)

**S3 Appendix. In-depth interview guide.** This is an in-depth interview guide we used to interview the participants in our study.
(DOCX)

**S1 File. Consolidated criteria for reporting qualitative studies (COREQ): 32-item checklist.**
(DOCX)

## Acknowledgments

We thank the Federal Ministry of Health and regional health bureaus of Amhara, Oromia, the Southern National, Nationalities, and Peoples' Region, and Tigray regions, for their support in implementing these surveys. We take this opportunity to extend our gratitude to all the study participants for the time they gave to respond to the survey questionnaires and provide us with valuable information. We would like to acknowledge our colleagues at The Last Ten Kilometers 2020 Project of JSI Research & Training Institute, Inc. for their contributions at all stages of implementing this work. Finally, researchers would like to acknowledge our colleagues Adey Abebe and Julie Ray for editing this manuscript.

## Author Contributions

**Conceptualization:** Gizachew Tadele Tiruneh, Nebreed Fesseha Zemichael, Wuleta Aklilu Betemariam, Ali Mehryar Karim.

**Data curation:** Gizachew Tadele Tiruneh, Nebreed Fesseha Zemichael, Wuleta Aklilu Betemariam, Ali Mehryar Karim.

**Formal analysis:** Wuleta Aklilu Betemariam, Ali Mehryar Karim.

**Funding acquisition:** Nebreed Fesseha Zemichael, Wuleta Aklilu Betemariam, Ali Mehryar Karim.

**Investigation:** Gizachew Tadele Tiruneh, Nebreed Fesseha Zemichael, Ali Mehryar Karim.

**Methodology:** Gizachew Tadele Tiruneh, Nebreed Fesseha Zemichael, Wuleta Aklilu Betemariam, Ali Mehryar Karim.

**Project administration:** Nebreed Fesseha Zemichael, Wuleta Aklilu Betemariam, Ali Mehryar Karim.

**Resources:** Nebreed Fesseha Zemichael, Wuleta Aklilu Betemariam.

**Software:** Gizachew Tadele Tiruneh, Ali Mehryar Karim.

**Supervision:** Ali Mehryar Karim.

**Validation:** Gizachew Tadele Tiruneh, Wuleta Aklilu Betemariam, Ali Mehryar Karim.

**Visualization:** Gizachew Tadele Tiruneh, Ali Mehryar Karim.

**Writing – original draft:** Gizachew Tadele Tiruneh.

**Writing – review & editing:** Gizachew Tadele Tiruneh, Nebreed Fesseha Zemichael, Wuleta Aklilu Betemariam, Ali Mehryar Karim.

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
