## [Decision Letter · Decision Letter 0]

19 Aug 2019

PONE-D-19-15140

Effectiveness of Participatory Community Solutions Strategy on Improving Household and Provider Health Care Behaviors and Practices: A Mixed Method Evaluation

PLOS ONE

Dear Mr Tiruneh,

Thank you for submitting your manuscript to PLOS ONE. After careful consideration, we feel that it has merit but does not fully meet PLOS ONE’s publication criteria as it currently stands. Therefore, we invite you to submit a revised version of the manuscript that addresses the points raised during the review process.

I found your article of great interest and commend you for the important work you are doing in Ethiopia. In order to avoid further delay, I have made the decision of Major Revision based on one external reviewer and my own assessment of the manuscript. In addition to Reviewer #1’s comments, please see my additional comments below.

We would appreciate receiving your revised manuscript by Oct 03 2019 11:59PM. To enhance the reproducibility of your results, we recommend that if applicable you deposit your laboratory protocols in protocols.io, where a protocol can be assigned its own identifier (DOI) such that it can be cited independently in the future. For instructions see: http://journals.plos.org/plosone/s/submission-guidelines#loc-laboratory-protocols

We look forward to receiving your revised manuscript.

Kind regards,

Emily A Hurley, M.P.H., Ph.D.

Academic Editor

PLOS ONE

Journal Requirements:

2. Data availability. Please note that authors should not be the sole named individuals responsible for ensuring data access. If these data cannot be publicly deposited or included in the supporting information, e.g. due to patient privacy, legal reasons, or being provided by a third party, please explain why and explain how researchers may access them via a named data access committee or named ethics committee.

3. Please specify in your financial disclosure whether the funders played any role in the study.  This information should be included in your cover letter; we will change the online submission form on your behalf.

4. Please specify whether an interview guide was used to interview the participants in your study. If yes, please describe and/or include a copy as a Supporting Information file.

5. Please include additional information regarding the survey used in the study and ensure that you have provided sufficient details that others could replicate the analyses. For instance, if you developed a questionnaire as part of this study and it is not under a copyright more restrictive than CC-BY, please include a copy, in both the original language and English, as Supporting Information.

6. Please carefully proof read your manuscript. For example, there is a missing space in the abstract “…November 2017.Propensity scores…”.

7. Please amend your current ethics statement to address the following concerns:   

a) Did participants provide their written or verbal informed consent to participate in this study?  

b) If consent was verbal, please explain:  

i) Why was written consent not obtained?   

ii) How did you record/document participant consent?  

iii) Did the ethics committees/IRBs approve this consent procedure?

8. Thank you for stating the following in the Competing Interests section:

We note that one or more of the authors are employed by a commercial company:  JSI Research & Training Institute, Inc

9.  Thank you for including your ethics statement:

"Ethical clearances for the surveys were obtained from the ethical review boards of Amhara, Oromia, SNNP, Tigray Regional Health Bureaus, and JSI. All the study participants were informed about the purpose of the study; benefits and hazards of the study were explained to all study participants, and each participant was notified of their right to opt out when responding to questions. Consent was sought before conducting any interviews. Moreover, the information obtained from the research participants were kept in private (codes were used during reporting of the IDI quotes)."

a. Please amend your current ethics statement to confirm that your named institutional review board or ethics committee specifically approved this study.

10. We note that you have included the phrase “data not shown” in your manuscript. Unfortunately, this does not meet our data sharing requirements. PLOS does not permit references to inaccessible data. We require that authors provide all relevant data within the paper, Supporting Information files, or in an acceptable, public repository. Please add a citation to support this phrase or upload the data that corresponds with these findings to a stable repository (such as Figshare or Dryad) and provide and URLs, DOIs, or accession numbers that may be used to access these data. Or, if the data are not a core part of the research being presented in your study, we ask that you remove the phrase that refers to these data.

11. We note that you have indicated that data from this study are available upon request. PLOS only allows data to be available upon request if there are legal or ethical restrictions on sharing data publicly. For more information on unacceptable data access restrictions, please see http://journals.plos.org/plosone/s/data-availability#loc-unacceptable-data-access-restrictions.

Additional Editor Comments (if provided):

I commend the authors for this important work and hope they can sufficiently address the comments from myself and Reviewer #1.

Major Comments:

The qualitative results are difficult to interpret given the lack of detail about the methods of data collection and analysis. Please give more detail about the qualitative portion of the study, referring to PLOS ONE guidelines below:

Qualitative research studies should be reported in accordance to the Consolidated criteria for reporting qualitative research (COREQ) checklist. Further reporting guidelines can be found in the Equator Network's Guidelines for reporting qualitative research

The qualitative section “improved service utilization and qualitative of care” does not seem to add much to the overall study. What value does perceived effectiveness add if there is quantitative data to more objectively assess effectiveness? How do we know the answers of the WDA to their funders were not influenced by social desirability bias?

I agree with Reviewer #1’s comment about the multiple outcomes. Was there a primary outcome that was chosen prior to the trail as the indicator of intervention success, or one that the intervention was most focused on achieving? Or, among the many indicators, was there any statistical adjustment for multiple outcomes?

The literature review could be more robust, especially when referencing evidenced-based intervention (e.g. kangaroo care), please take care to add appropriate citations.

Minor Comments:

Abstract

Please edit for grammar: line 21 “We evaluated the effects of this strategy using a mixed methods research” and again in line 80 (take out “a” or rephrase to: “a mixed-methods research strategy)

Line 25 A general reader would not know what is meant by “kebeles”. Please define or reword

Intro

Grammar: line 45 “there is high fall out rates”

Table 2

Readiness to perform signal functions

- Please indicate if the “availability of…” indicators use a dichotomous (yes/no) response, or if there is some other measure of levels of availability

Lines 238-248: Please justify your choice of these variables as potential confounders, and if appropriate add references

Please justify with a reference or other explanation the choice of a 0.2 p-value for the logit stepwise backward selection

Line numbers stop on page 31.

There appears to be a missing quotation in the middle of page 34

Middle of page 36: Please give some more detail about the “previous study in Ethiopia” and how the results of the present study align with it.

Middle of page 38: “synthesized to unpack” sounds contradictory

Throughout:

Recommend italicizing “woreda” and other local terms

Be consistent in capitalization for Family Conversation and Birth Notification

Reviewers' comments:

Reviewer's Responses to Questions

**Comments to the Author**

1. Is the manuscript technically sound, and do the data support the conclusions?

Reviewer #1: Partly

2. Has the statistical analysis been performed appropriately and rigorously? 

Reviewer #1: Yes

3. Have the authors made all data underlying the findings in their manuscript fully available?

Reviewer #1: No

4. Is the manuscript presented in an intelligible fashion and written in standard English?

Reviewer #1: No

5. Review Comments to the Author

Reviewer #1: I commend the authors on synthesising results of this important evaluation of a complex intervention strategy in multiple regions of Ethiopia. I wish to offer the following substantive and minor comments.

Substantive comments:

1. This is an evaluation of a complex intervention with multiple components. It would be helpful to include a table with all intervention components, their timeframes and geographical coverage in the methods section.

2. Strengthen and move qualitative sections:

In this article, the qualitative sections are used to: (a) help describe the interventions; (b) highlight barriers and facilitators to implementation; and (c) derive lessons for future implementation. Unfortunately all this is done without presenting any actual qualitative data. I would recommend simplifying this and using a table or diagramme to present the content of interventions (the innovation and implementation mechanisms, line 287 onwards) and removing the sections where qualitative data on intervention implementation are summarised. I would also recommend condensing sections on lessons learned or providing data to support the statements made there. The most valuable use of the qualitative data is in the section on barriers and facilitators. This can be strengthened through the use of quotes and reporting of discrepant cases, otherwise they is no actual qualitative analysis, just a brief summary of views from programme insiders.

Consider moving remaining qualitative sections (barriers and facilitators) after the DID results so they help explain your results. If you do not feel like you can do the above, I would recommend publishing the qualitative analysis in a separate article where you can really do it justice.

Finally, I would remove the reference to themes 'emerging' from the data as it seems that data were mostly coded in response to the research questions.

3. Background - You could state what the quantitative research questions were/was, for balance. At the moment you also list the qualitative questions.

4. Was the impact evaluation registered? If so where is this registration number?

5. The quantitative impact evaluation has multiple outcomes, was there an a priori data analysis plan, and was any adjustment made for multiple hypothesis testing?

7. Abstract and Results - Your data do not support that "receiving early antenatal care between baseline and follow-up surveys in the intervention area is attributable to the strategy" because the confidence interval includes 0, i.e. the possibility of no effect. This needs to be removed.

8. Discussion - the main reason for lack of effect on most MNCH indicators aside from skilled care at birth and receipt of PNC for mother and baby was the minimal intervention duration time. Does anything else explain the results?

9. Discussion - A key limitation of the qualitative work is the lack of sampling community members not linked to the project, i.e. non-project-based staff, HEWs or HDAs. Is there a reason why you did not speak with mothers about their views of what changed and what did not?

10. For your consideration - I would avoid the use of the term 'innovations' - community interventions like family conversations and facility or health systems-levels interventions like QI or PDSA cycles are not really 'innovations'. They may be in this context, but there is a long history of their use in multiple contexts. The donor-favoured term 'innovation' is a little tiring; can we just call an intervention an intervention?

11. Conflict of interest: the senior author of this article is from the Gates Foundation, which also funded the writing of this article and, critically, the interventions being evaluated. The role of Foundation-paid staff in writing up/editing results from their own projects needs to be explained or declared more clearly as a conflict of interest. Please see recent demands for transparency in such situations: https://gh.bmj.com/content/4/3/e001746

I wish you good luck for the revisions and commend you on such hard and important work.

6. PLOS authors have the option to publish the peer review history of their article (what does this mean?). If published, this will include your full peer review and any attached files.

Reviewer #1: No

---

## [Author Response · Author response to Decision Letter 0]

1 Oct 2019

Point-by-point response to reviewer/editorial

Version 1

Journal: PLOS ONE 

Title: " Effectiveness of Participatory Community Solutions Strategy on Improving Household and Provider Health Care Behaviors and Practices: A Mixed Method Evaluation (PONE-D-19-15140)”

The authors would like to appreciate and thank the reviewers for the constructive comments.

Our point-by-point responses to the reviewers are below each of the comments in italics. We also make sure that this version of the manuscript conforms the journal style. 

Journal Requirements:

Comment well taken. 

2. Data availability. Please note that authors should not be the sole named individuals responsible for ensuring data access. If these data cannot be publicly deposited or included in the supporting information, e.g. due to patient privacy, legal reasons, or being provided by a third party, please explain why and explain how researchers may access them via a named data access committee or named ethics committee.

Comment well taken. And in this version, the data used for this analysis are submitted as supplementary material, S1 Appendix.

3. Please specify in your financial disclosure whether the funders played any role in the study. This information should be included in your cover letter; we will change the online submission form on your behalf.

Thanks for reminding us. The following paragraph included in the financial disclosure section. “The article write-up and publication fee was supported by the Bill & Melinda Gates Foundation, Grant Number OPP1131042. JSI Research & Training Institute, Inc. has provided us support in the form of salaries for authors [GT, AK, NZ, WB]. However, any of the funders did not have role in study design, data collection and analysis, decision to publish, or preparation of the manuscript.”

4. Please specify whether an interview guide was used to interview the participants in your study. If yes, please describe and/or include a copy as a Supporting Information file.

Comment well taken. Interview guides were used to capture the data and in this version, interview guides are included as supplementary files, S3 Appendix.

5. Please include additional information regarding the survey used in the study and ensure that you have provided sufficient details that others could replicate the analyses. For instance, if you developed a questionnaire as part of this study and it is not under a copyright more restrictive than CC-BY, please include a copy, in both the original language and English, as Supporting Information.

Comment well taken. Standard questionnaire was used to capture the data and in this version, the questionnaire is included as supplementary files, S2 Appendix.

6. Please carefully proof read your manuscript. For example, there is a missing space in the abstract “…November 2017.Propensity scores…”.

Well noted.

7. Please amend your current ethics statement to address the following concerns: 

a) Did participants provide their written or verbal informed consent to participate in this study? 

b) If consent was verbal, please explain: 

i) Why was written consent not obtained? 

ii) How did you record/document participant consent? 

iii) Did the ethics committees/IRBs approve this consent procedure?

The ethics statement now addressed the issues raised and it reads as follows; “Ethical clearances for the surveys were obtained from the ethical review boards of Amhara, Oromia, SNNP, Tigray Regional Health Bureaus, and JSI. All the study participants were informed about the purpose of the study; benefits and hazards of the study were explained to all study participants, and each participant was notified of their right to opt out when responding to questions. Verbal Consent was sought and documented before conducting any interviews. If the respondent was less than 18 years old, then consent was sought from her husband or guardian. Majority of the respondents were not expected to be able to read or write; as such, written consent was not sought. If the respondent agreed to be interviewed after listening to the consent statement, the interviewer marked the questionnaire as consent given below the consent statement and then signed below that. The interviewer continued with the interview only after receiving and documenting the consent. The survey protocol submitted to the ethical review committee included the study questionnaire with the consent statement that described the consent obtaining procedure and it was approved by the committee. Moreover, the information obtained from the research participants were kept in private (codes were used during reporting of the IDI quotes).”

8. Thank you for stating the following in the Competing Interests section:

We note that one or more of the authors are employed by a commercial company: JSI Research & Training Institute, Inc

Comment well taken. And the Funding Statement and Competing Interests Statement is updated as follows: “The article write-up and publication fee was supported by the Bill & Melinda Gates Foundation, Grant Number OPP1131042. The authors have been working for JSI Research & Training Institute, Inc., a commercial company. We declared that this commercial affiliation does not alter our adherence to PLOS ONE policies on sharing data and materials. JSI Research & Training Institute, Inc. has provided us support in the form of salaries for authors [GT, AK, NZ, WB]. However, any of the funders did not have role in study design, data collection and analysis, decision to publish, or preparation of the manuscript.”

9. Thank you for including your ethics statement:

"Ethical clearances for the surveys were obtained from the ethical review boards of Amhara, Oromia, SNNP, Tigray Regional Health Bureaus, and JSI. All the study participants were informed about the purpose of the study; benefits and hazards of the study were explained to all study participants, and each participant was notified of their right to opt out when responding to questions. Consent was sought before conducting any interviews. Moreover, the information obtained from the research participants were kept in private (codes were used during reporting of the IDI quotes)."

a. Please amend your current ethics statement to confirm that your named institutional review board or ethics committee specifically approved this study.

The ethics statement now amended as follows; “Ethical clearances for the surveys were obtained from the ethical review boards of Amhara, Oromia, SNNP, Tigray Regional Health Bureaus, and JSI. All the study participants were informed about the purpose of the study; benefits and hazards of the study were explained to all study participants, and each participant was notified of their right to opt out when responding to questions. Verbal Consent was sought and documented before conducting any interviews. If the respondent was less than 18 years old, then consent was sought from her husband or guardian. Majority of the respondents were not expected to be able to read or write; as such, written consent was not sought. If the respondent agreed to be interviewed after listening to the consent statement, the interviewer marked the questionnaire as consent given below the consent statement and then signed below that. The interviewer continued with the interview only after receiving and documenting the consent. The survey protocol submitted to the ethical review committee included the study questionnaire with the consent statement that described the consent obtaining procedure and it was approved by the committee. Moreover, the information obtained from the research participants were kept in private (codes were used during reporting of the IDI quotes).”

10. We note that you have included the phrase “data not shown” in your manuscript. Unfortunately, this does not meet our data sharing requirements. PLOS does not permit references to inaccessible data. We require that authors provide all relevant data within the paper, Supporting Information files, or in an acceptable, public repository. Please add a citation to support this phrase or upload the data that corresponds with these findings to a stable repository (such as Figshare or Dryad) and provide and URLs, DOIs, or accession numbers that may be used to access these data. Or, if the data are not a core part of the research being presented in your study, we ask that you remove the phrase that refers to these data.

 Comment well taken. And Table is inserted as source in the text.

11. We note that you have indicated that data from this study are available upon request. PLOS only allows data to be available upon request if there are legal or ethical restrictions on sharing data publicly. For more information on unacceptable data access restrictions, please see http://journals.plos.org/plosone/s/data-availability#loc-unacceptable-data-access-restrictions.

The survey dataset we used for the analysis is provided as supplementary information (S1 Appendix).

Additional Editor Comments (if provided):

I commend the authors for this important work and hope they can sufficiently address the comments from myself and Reviewer #1.

Major Comments:

The qualitative results are difficult to interpret given the lack of detail about the methods of data collection and analysis. Please give more detail about the qualitative portion of the study, referring to PLOS ONE guidelines below:

Qualitative research studies should be reported in accordance to the Consolidated criteria for reporting qualitative research (COREQ) checklist. Further reporting guidelines can be found in the Equator Network's Guidelines for reporting qualitative research

The qualitative section “improved service utilization and qualitative of care” does not seem to add much to the overall study. What value does perceived effectiveness add if there is quantitative data to more objectively assess effectiveness? How do we know the answers of the WDA to their funders were not influenced by social desirability bias?

Comment well taken. The qualitative sections are moved as suggested by Reviewer #1 and the COREQ checklist is followed. Moreover, the possibility of recall and social-desirability bias are presented as limitations of this paper in the Discussion section. 

I agree with Reviewer #1’s comment about the multiple outcomes. Was there a primary outcome that was chosen prior to the trail as the indicator of intervention success, or one that the intervention was most focused on achieving? Or, among the many indicators, was there any statistical adjustment for multiple outcomes?

The project was designed to influence multiple RMNCH indicators including antepartum, intrapartum, and postpartum. As, it was clearly indicated in the project document as well as the research protocol, below are the list of indicators the project intended to influence; 

Antenatal care: antenatal care (ANC); complete ANC (BP measured and urine & blood tested); ANC in 1st trimester; 4 or more ANC visits (ANC 4+), ANC in 1st & last trimester

Perinatal care: Institutional delivery/skilled health personnel; institutional delivery among the lowest wealth quintile; MgSO4 given for pre-eclampsia; Oxytocin given during 3rd stage of labor; partograph use rate; met need for emergency obstetric & newborn care (EmONC); signal functions of basic EmONC (BEmONC); still birth rates. 

Postnatal & newborn care: newborn care, neonatal infection; neonatal asphyxia managed; postnatal care (PNC) in 48 hours; complete PNC in 48 hours. 

As such, there was no adjustment made for multiple hypothesis testing.

The literature review could be more robust, especially when referencing evidenced-based intervention (e.g. kangaroo care), please take care to add appropriate citations.

Comment well acknowledged. Availability of evidence-based, low-cost, life-saving MNH interventions as well as their impact and implementation challenges are now presented in the Introduction section, paragraph 1 and 2 of this version. 

Minor Comments:

Abstract

Please edit for grammar: line 21 “We evaluated the effects of this strategy using a mixed methods research” and again in line 80 (take out “a” or rephrase to: “a mixed-methods research strategy)

Line 25 A general reader would not know what is meant by “kebeles”. Please define or reword

Thanks for the comments. Now, it is corrected

Intro

Grammar: line 45 “there is high fall out rates”

Thanks for the comments. Now, it is corrected

Table 2

Readiness to perform signal functions

- Please indicate if the “availability of…” indicators use a dichotomous (yes/no) response, or if there is some other measure of levels of availability

Comments well acknowledged and indicated as yes/no response. 

Lines 238-248: Please justify your choice of these variables as potential confounders, and if appropriate add references

Please justify with a reference or other explanation the choice of a 0.2 p-value for the logit stepwise backward selection

Potential confounders are identified from previous studies and these variables are indicated in the Introduction section of the manuscript, Page 4, line 54-59.

Thanks a lot for the comments. Scholars recommended to match variables on the logit of the propensity score using calipers of width equal to 0.2 of the standard deviation of the logit of the propensity score when estimating differences in means or risk differences. Relevant references are now cited (Austin, 2009, 2011)., page 19, line 300.

Line numbers stop on page 31.

Thanks for the comments. Now we make the line number continuous.

There appears to be a missing quotation in the middle of page 34

Sure, it was missed. Now, inserted.

Middle of page 36: Please give some more detail about the “previous study in Ethiopia” and how the results of the present study align with it.

Well acknowledged and addressed as suggested. 

Middle of page 38: “synthesized to unpack” sounds contradictory

Noted and corrected 

Throughout:

Recommend italicizing “woreda” and other local terms

Be consistent in capitalization for Family Conversation and Birth Notification

Comments well taken

Comments to the Author

1. Is the manuscript technically sound, and do the data support the conclusions?

Reviewer #1: Partly

 Major revisions made on the data presentation, particularly the qualitative section.

2. Has the statistical analysis been performed appropriately and rigorously? 

Reviewer #1: Yes

 3. Have the authors made all data underlying the findings in their manuscript fully available?

Reviewer #1: No

 Now included as supplementary file, S1 Appendix.

4. Is the manuscript presented in an intelligible fashion and written in standard English?

Reviewer #1: No

We improved the language in this version.

5. Review Comments to the Author

Reviewer #1: I commend the authors on synthesising results of this important evaluation of a complex intervention strategy in multiple regions of Ethiopia. I wish to offer the following substantive and minor comments.

Substantive comments:

1. This is an evaluation of a complex intervention with multiple components. It would be helpful to include a table with all intervention components, their timeframes and geographical coverage in the methods section.

Comment well acknowledged. Table 1 is inserted in this version. 

2. Strengthen and move qualitative sections:

In this article, the qualitative sections are used to: (a) help describe the interventions; (b) highlight barriers and facilitators to implementation; and (c) derive lessons for future implementation. Unfortunately all this is done without presenting any actual qualitative data. I would recommend simplifying this and using a table or diagramme to present the content of interventions (the innovation and implementation mechanisms, line 287 onwards) and removing the sections where qualitative data on intervention implementation are summarised. I would also recommend condensing sections on lessons learned or providing data to support the statements made there. The most valuable use of the qualitative data is in the section on barriers and facilitators. This can be strengthened through the use of quotes and reporting of discrepant cases, otherwise they is no actual qualitative analysis, just a brief summary of views from programme insiders.

Consider moving remaining qualitative sections (barriers and facilitators) after the DID results so they help explain your results. If you do not feel like you can do the above, I would recommend publishing the qualitative analysis in a separate article where you can really do it justice.

Finally, I would remove the reference to themes 'emerging' from the data as it seems that data were mostly coded in response to the research questions.

Comment well taken. Detailed description of the intervention is now presented in Table 1. Moreover, the qualitative sections are moved as suggested. More quotes are added to strengthened the evidence of the findings.

3. Background - You could state what the quantitative research questions were/was, for balance. At the moment you also list the qualitative questions.

Comment well taken. We convinced that objectives can clearly convey the intended message. As such, in this version, only objectives of both the qualitative and quantitative studies are presented and for balance, we dropped the research questions of the qualitative study. 

4. Was the impact evaluation registered? If so where is this registration number?

This evaluation was indicated in the project document. Following the project design, a research protocol was developed. The evaluation protocol was not registered in any of the online registries. The authors can submit a draft protocol as supplementary material, if needed.

5. The quantitative impact evaluation has multiple outcomes, was there an a priori data analysis plan, and was any adjustment made for multiple hypothesis testing?

The project was designed to influence the following antepartum, intrapartum, and postpartum indicators: 

Antenatal care: antenatal care (ANC); complete ANC (BP measured and urine & blood tested); ANC in 1st trimester; 4 or more ANC visits (ANC 4+), ANC in 1st & last trimester, tested for Syphilis. 

Perinatal care: Institutional delivery; skilled birth attendance; MgSO4 given for pre-eclampsia; prophylactic uterotonics; antibiotics for pPRom; antenatal corticosteroids; partograph use rate; met need for emergency obstetric & newborn care (EmONC); signal functions of basic EmONC (BEmONC); still birth rates. 

Postnatal & newborn care: Immediate drying the baby; immediate breastfeeding; kangaroo mother care of facility deliveries; safe cord care; thermal care; neonatal resuscitation; neonatal infection management; maternal postnatal care (PNC); complete maternal PNC; neonatal postnatal care (PNC); complete neonatal PNC; FP counseling provided during PNC. 

This evaluation was based on the project document. As such, there was no any adjustment made to test multiple hypothesis after or before data collection.

7. Abstract and Results - Your data do not support that "receiving early antenatal care between baseline and follow-up surveys in the intervention area is attributable to the strategy" because the confidence interval includes 0, i.e. the possibility of no effect. This needs to be removed.

Comment well taken.

8. Discussion - the main reason for lack of effect on most MNCH indicators aside from skilled care at birth and receipt of PNC for mother and baby was the minimal intervention duration time. Does anything else explain the results?

Comment well take. And in this version, we supplemented the possible reasons from the qualitative study for low intervention coverage. Page 43, line 706-9.

9. Discussion - A key limitation of the qualitative work is the lack of sampling community members not linked to the project, i.e. non-project-based staff, HEWs or HDAs. Is there a reason why you did not speak with mothers about their views of what changed and what did not?

The main objective of the qualitative study was to describe the process of implementation and understand the complex participatory QI process; highlight barriers and facilitators of the implementation and scalability of the intervention. We believe active community level intervention players, community volunteers (WDAs) and front line health workers (HEWs) would be the key informants for the purpose. Accordingly, we did not interview mothers, direct beneficiaries for the project.

10. For your consideration - I would avoid the use of the term 'innovations' - community interventions like family conversations and facility or health systems-levels interventions like QI or PDSA cycles are not really 'innovations'. They may be in this context, but there is a long history of their use in multiple contexts. The donor-favoured term 'innovation' is a little tiring; can we just call an intervention an intervention?

Comment well taken. 

11. Conflict of interest: the senior author of this article is from the Gates Foundation, which also funded the writing of this article and, critically, the interventions being evaluated. The role of Foundation-paid staff in writing up/editing results from their own projects needs to be explained or declared more clearly as a conflict of interest. Please see recent demands for transparency in such situations: https://gh.bmj.com/content/4/3/e001746

The senior author of this manuscript has been working for JSI Research and Training Institute Inc. during the design, field work, and report-up of this manuscript. He recently left JSI and joined Gates Foundation, In February 2019. 

We updated the Funding Statement and Competing Interests Statement as follows: “The article write-up and publication fee was supported by the Bill & Melinda Gates Foundation, Grant Number OPP1131042. The authors have been working for JSI Research & Training Institute, Inc., a commercial company. We declared that this commercial affiliation does not alter our adherence to PLOS ONE policies on sharing data and materials. JSI Research & Training Institute, Inc. has provided us support in the form of salaries for authors [GT, AK, NZ, WB]. However, any of the funders did not have role in study design, data collection and analysis, decision to publish, or preparation of the manuscript.”

---

## [Editor Report · Decision Letter 1]

27 Nov 2019

PONE-D-19-15140R1

Effectiveness of Participatory Community Solutions Strategy on Improving Household and Provider Health Care Behaviors and Practices: A Mixed-Method Evaluation

PLOS ONE

Dear Mr Tiruneh,

Thank you for submitting your manuscript to PLOS ONE. After careful consideration, we feel that it has merit but does not fully meet PLOS ONE’s publication criteria as it currently stands. Therefore, we invite you to submit a revised version of the manuscript that addresses the points raised during the review process.

We apologize for the delay in returning this. I could not identify additional reviewers for this round, but have reviewed your work and provided my own review. Thank you for taking the time to carefully address the comments on the original submission. I do think this version is much improved, but do ask that you attend to the editorial comments below.

We would appreciate receiving your revised manuscript by Jan 11 2020 11:59PM. To enhance the reproducibility of your results, we recommend that if applicable you deposit your laboratory protocols in protocols.io, where a protocol can be assigned its own identifier (DOI) such that it can be cited independently in the future. For instructions see: http://journals.plos.org/plosone/s/submission-guidelines#loc-laboratory-protocols

We look forward to receiving your revised manuscript.

Kind regards,

Emily A Hurley, M.P.H., Ph.D.

Academic Editor

PLOS ONE

Additional Editor Comments (if provided):

Intro

Third line- “being” can be deleted, can be simply “care provided”

First line of second paragraph: specify what “it” is

Last paragraph of the intro could be improved. In the line proceeding, you indicate that the study was mixed-methods, and then only describe the qualitative component in the next paragraph. Be clear as if you are presenting results from the quantitative sections or not. If so, state the overall objective of the mixed-methods study as well as the quantitative and qualitive sections.

Methods

More detail on the qualitative analysis methods are needed. I once again recommend the COREQ checklist to ensure you report on all the necessary criteria for qualitative studies, particularly the data analysis section. How many coders? How were themes derived? What did the coding tree entail? (etc..)

Results

The first sentence of this section needs editing, as the results are not longer presented in four sections.

It does seem like with so many different statistical tests, adjusting the p-value for multiple comparisons is warranted, particularly when the authors place so much emphasis on the 0.05 value as a cut-off for significance. Please adjust for these multiple comparisons or provide more detail as to why this is not necessary in your current study.

For the qualitative quotes please cut the participant identifiers (e.g. HDA33235) or explain their significance. As they stand now they are difficult to interpret.

Please edit this section for grammar as well e.g. there is a sentence fragment “participants in Amhara (IDI11413)” after the first quote of the “enhanced the knowledge and skills of health workers” section. There are also many references to “participants” as plural after quotes- were all of these quotes said by more than one person?

Discussion

There are still a number of typos. Please proofread carefully (e.g. “chanege" should be “change” in 5th paragraph of discussion)

Please define “support system” as you refer to it in this section. From the paragraph, it looks like you are referring to a strong network of implementing partners, but please be specific, as “support system” could take many different meanings.

---

## [Author Response · Author response to Decision Letter 1]

8 Jan 2020

Point-by-point response to reviewer/editorial

Version 2

Journal: PLOS ONE 

Title: " Effectiveness of Participatory Community Solutions Strategy on Improving Household and Provider Health Care Behaviors and Practices: A Mixed Method Evaluation (PONE-D-19-15140R1)”

The authors would like to appreciate and thank the editor for the constructive comments.

Our point-by-point responses are below each of the comments in italics. We also make sure that this version of the manuscript conforms the journal style. The manuscript is reviewed by a native speaker, Julie Ray.

Intro

Third line- “being” can be deleted, can be simply “care provided”

First line of second paragraph: specify what “it” is

Thank you so much for the comments. The language is now edited with a native speaker and grammatically improved. 

Last paragraph of the intro could be improved. In the line proceeding, you indicate that the study was mixed-methods, and then only describe the qualitative component in the next paragraph. Be clear as if you are presenting results from the quantitative sections or not. If so, state the overall objective of the mixed-methods study as well as the quantitative and qualitive sections.

Thanks a lot for the valuable comments. The quantitative component is now described (line 94-96, page 6).

Methods

More detail on the qualitative analysis methods are needed. I once again recommend the COREQ checklist to ensure you report on all the necessary criteria for qualitative studies, particularly the data analysis section. How many coders? How were themes derived? What did the coding tree entail? (etc..)

Thank you so much for the valid comments. Details are included regarding data collectors, saturation, analysis (coding and theme), ﬁndings, and reporting (line 242-63, page 15 and line 313-20, page 20).

Results

The first sentence of this section needs editing, as the results are not longer presented in four sections.

Comment well taken. This section is now edited and presented on page 20 (line 322-4) and page 29 (line 404-8).

It does seem like with so many different statistical tests, adjusting the p-value for multiple comparisons is warranted, particularly when the authors place so much emphasis on the 0.05 value as a cut-off for significance. Please adjust for these multiple comparisons or provide more detail as to why this is not necessary in your current study.

Thank you so much for the valuable comments; comments are well acknowledged. Adjusting the p-values for multiple hypotheses tests is not always necessary. Since we report the p-values, the readers have the opportunity to make adjustments if they feel necessary. To address the issue, we added the following to the discussion section, page 36, line 561-5.

 “Adjusting p-values is often recommended when conducting multiple hypotheses tests `simultaneously’ [1-4]. However, it is not always necessary [5-7]. As such, we did not adjust the p-value in our case as it would have increased the probability of making type II error, i.e., concluding that the intervention was not effective when it is true [8, 5, 6, 9].”

For the qualitative quotes please cut the participant identifiers (e.g. HDA33235) or explain their significance. As they stand now they are difficult to interpret.

Please edit this section for grammar as well e.g. there is a sentence fragment “participants in Amhara (IDI11413)” after the first quote of the “enhanced the knowledge and skills of health workers” section. There are also many references to “participants” as plural after quotes- were all of these quotes said by more than one person?

Comments are well taken and all participant identifiers are defined and grammatical errors are fixed. 

Discussion

There are still a number of typos. Please proofread carefully (e.g. “chanege" should be “change” in 5th paragraph of discussion)

Thank you so much for the comments. The language is now edited with a native speaker and grammatically improved. 

Please define “support system” as you refer to it in this section. From the paragraph, it looks like you are referring to a strong network of implementing partners, but please be specific, as “support system” could take many different meanings.

Thanks for the valuable comment. It is now defined (line 553-4, page 36)

References

1. Tukey JW. Some thoughts on clinical trials, especially problems of multiplicity. Science. 1977;198(4318):679-84. 

2. Bland JM, Altman DG. Multiple significance tests: the Bonferroni method. Bmj. 1995;310(6973):170. 

3. Greenhalgh T. How to read a paper: Statistics for the non-statistician. I: Different types of data need different statistical tests. Bmj. 1997;315(7104):364-6. 

4. Ludbrook J. Multiple comparison procedures updated. Clinical and Experimental Pharmacology and Physiology. 1998;25(12):1032-7. 

5. Feise RJ. Do multiple outcome measures require p-value adjustment? BMC medical research methodology. 2002;2(1):8. 

6. Perneger TV. What's wrong with Bonferroni adjustments. Bmj. 1998;316(7139):1236-8. 

7. Rothman KJ. No adjustments are needed for multiple comparisons. Epidemiology. 1990:43-6. 

8. Cole P. The evolving case-control study. The Case-Control Study Consensus and Controversy. Elsevier; 1979. p. 15-27.

9. Thomas D, Siemiatycki J, Dewar R, Robins J, Goldberg M, Armstrong B. The problem of multiple inference in studies designed to generate hypotheses. Am J Epidemiol. 1985;122(6):1080-95.

---

## [Editor Report · Decision Letter 2]

9 Jan 2020

Effectiveness of Participatory Community Solutions Strategy on Improving Household and Provider Health Care Behaviors and Practices: A Mixed-Method Evaluation

PONE-D-19-15140R2

Dear Dr. Tiruneh,

We are pleased to inform you that your manuscript has been judged scientifically suitable for publication and will be formally accepted for publication once it complies with all outstanding technical requirements.

With kind regards,

Emily A Hurley, M.P.H., Ph.D.

Academic Editor

PLOS ONE
---

## [Editor Report · Acceptance letter]

15 Jan 2020

PONE-D-19-15140R2 

Effectiveness of Participatory Community Solutions Strategy on Improving Household and Provider Health Care Behaviors and Practices: A Mixed-Method Evaluation 

Dear Dr. Tiruneh:

I am pleased to inform you that your manuscript has been deemed suitable for publication in PLOS ONE. Congratulations! Your manuscript is now with our production department. 

With kind regards,

on behalf of

Dr. Emily A Hurley 

Academic Editor

PLOS ONE